# CLUE: Conflict-guided Localization for LLM Unlearning Framework

**Hang Chen**
School of Computer Science and Technology
Xi'an Jiaotong University
`albert2123@stu.xjtu.edu.cn`

**Jiaying Zhu**
School of Computer Science and Engineering
The Chinese University of Hong Kong
`jyzhu24@cse.cuhk.edu.hk`

**Xinyu Yang**[*]
School of Computer Science and Technology
Xi'an Jiaotong University
`yxyphd@mail.xjtu.edu.cn`

**Wenya Wang**[†]
School of Computer Science and Engineering
Nanyang Technological University
`wangwy@ntu.edu.sg`

## Abstract

The LLM unlearning aims to eliminate the influence of undesirable data without affecting causally unrelated information. This process typically involves using a **forget set** to remove target information, alongside a **retain set** to maintain non-target capabilities. While recent localization-based methods demonstrate promise in identifying important nodes (neurons) to be unlearned, they fail to disentangle nodes responsible for forgetting undesirable knowledge or retaining essential skills, often treating them as a single entangled group. As a result, these methods apply uniform interventions, risking catastrophic over-forgetting or incomplete erasure of the target knowledge. To address this, we turn to circuit discovery, a mechanistic interpretability technique, and propose the **C**onflict-guided **L**ocalization for LLM **U**nlearning fram**E**work (**CLUE**). This framework identifies the forget and retain circuit composed of important nodes, and then the circuits are transformed into conjunctive normal forms (CNF). The assignment of each node in the CNF satisfiability solution reveals whether it should be forgotten or retained. We then provide targeted fine-tuning strategies for different categories of nodes. Extensive experiments demonstrate that, compared to existing localization methods, CLUE achieves superior forget efficacy and retain utility through precise neural localization. Our code is available at `https://github.com/Zodiark-ch/CLUE`.

## 1 Introduction

Large language model (LLM) unlearning (Liu et al., 2025; Yao et al., 2024), as a machine learning method inherited from model unlearning (Cao & Yang, 2015; Neel et al., 2021), aims to have the LLM avoid or remove certain target information while preserving its other non-target capabilities as much as possible. Formally, the framework of LLM unlearning typically involves two datasets: the **forget set** and the **retain set**. The optimization objective is to avoid the original responses to the forget set (which are typically harmful or sensitive) and retain the existing responses to the retain set.

Many categories of methods currently exist for LLM unlearning (Zhang et al., 2024; Jia et al., 2023; Liu et al., 2024). Among these, **localization-informed unlearning** offers better interpretability by localizing key nodes (neurons)[1] or parameters. This also allows for better maintenance and targeted updates to these parameters, which aligns well with future modular machine learning developments (Menik & Ramaswamy, 2023). Recently, various localization-informed methods

---

[*]Corresponding author

[†]Corresponding author

[1]In this paper, the terms "node" and "neuron" represent almost the same concept. They are all used to represent some kind of trainable parameter matrix in the model (see Sec 2.2 for details). For consistency, and without loss of generality, we use "nodes" by default.

have emerged, including those based on gradients (Wu et al., 2023; Yu et al., 2023), weight attribution (Jia et al., 2024), and causal effect estimation (Patil et al., 2023; Meng et al., 2022).

However, existing localization methods can only identify an entangled set of "important" nodes resulting from the joint optimization on the forget and retain sets. They cannot pinpoint the specific subset of nodes responsible for forgetting, retaining, or a combination of both. As depicted in Figure 1, these important nodes can be intuitively subdivided into three categories: **retain nodes**, which influence only the retain set; **forget nodes**, which influence only the forget set; and **conflict nodes**, which influence both the forget and retain sets.

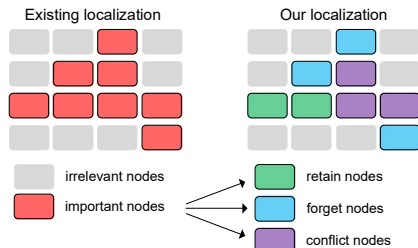

Figure 1: A fine-grained exploration inspired by existing localization methods.

Clearly separating these three types of nodes is essential for improving unlearning's **forget efficacy** and **retain utility**. For instance, suppose the forget set contains harmful information, whereas the retain set includes sentiment recognition samples. A coarse-grained intervention on all "important nodes" might unintentionally compromise the model's sentiment recognition capability when removing the harmful information. By disentangling the retain nodes from the "important nodes", the sentiment recognition capability is better preserved. However, disentangling these node types from the coarser set of important nodes is highly challenging. For example, in gradient-based methods, the gradient from the joint optimization of the forget loss and retain loss is not equivalent to the linear combination of the gradients from optimizing each loss separately. Therefore, the identified nodes reflect intertwined signals and cannot distinctly represent forgetting or retaining.

To address this problem, we resort to **circuit discovery** (Conmy et al., 2023; Bhaskar et al., 2024), a mechanistic interpretability method that identifies the important nodes and their activation relationships for a target task or dataset, representing them in a graph-structured circuit. This method enables explicit tracking of information flows between nodes, allowing us to isolate functional substructures. Recent circuit discovery techniques (Chen et al., 2025; Heimersheim & Nanda, 2024) are also capable of uncovering the logical relationships within a circuit; for example, some subcircuits resemble digital logic gates like an AND gate, while others are similar to an OR gate. This is particularly appealing in our setting, since forgetting and retaining are inherently compositional operations: for instance, a capability could only be preserved if multiple nodes are unchanged (similar to an AND gate), while another capability might be forgotten when just one of a group of nodes is edited (similar to an OR gate).

Building on this, we propose the **C**onflict-guided **L**ocalization for LLM **U**nlearning fram**E**work (**CLUE**), which can distinguish the node categories conceptualized in Figure 1 through the Boolean satisfiability of circuits. Specifically, CLUE first extracts circuits from the forget set and retain set, respectively. These circuits are then converted into Conjunctive Normal Form (CNF) using Tseitin's transformation (Tseitin, 1983). A CNF is constructed whose logic ensures that the forget circuit is modified while the retain circuit is preserved. By solving this CNF as a satisfiability problem (Fleury & Heisinger, 2020), we can determine which nodes belong to the retain node, forget node, and conflict node categories based on the values of each node in the optimal solution. Finally, we also provide a fine-tuning paradigm that provides supervision for the forget nodes using the forget loss and for the conflict nodes using both the forget and retain loss.

CLUE allows for the editing of different node categories based on distinct optimization objectives. This more fine-grained and precise localization enhances the effectiveness of unlearning. To demonstrate this, we conducted extensive experiments on three mainstream unlearning datasets: WMDP Cyber, WMDP Bio, and PKU SafeRLHF. The results not only show that our method significantly improves forget efficacy and retain utility with fewer modified parameters, but also reveal a more detailed correlation between the unlearning effect and the underlying circuits and nodes. In summary, our contributions are threefold:

- We propose conflict-based localization, which identifies more fine-grained node categories by solving the CNF satisfiability of the forget circuit and the retain circuit.

- We introduce CLUE, an effective framework for LLM unlearning that explicitly leverages different node categories to achieve more precise unlearning.

- We demonstrate through extensive experiments that CLUE surpasses current localization methods across multiple dimensions and offers more comprehensive interpretability.

## 2 PRELIMINARIES

### 2.1 LLM UNLEARNING

In light of the existing literature on LLM unlearning (Li et al., 2024; Maini et al., 2024; Ishibashi & Shimodaira, 2023; Yao et al., 2024; Pawelczyk et al., 2024), we define the problem of LLM unlearning as eliminating the influence of specific "unlearning targets" and removing associated model capabilities while preserving model performance for non-targets. To facilitate comprehension, we provide a commonly-used formulation of LLM unlearning problems below.

$$\min_{\theta} \mathbb{E}_{(x,y_f)\in\mathcal{D}_f}[\mathcal{L}(y_f|x;\theta)] + \lambda\mathbb{E}_{(x,y)\in\mathcal{D}_r}[\mathcal{L}(y|x;\theta)] \tag{1}$$

where $\lambda \geq 0$ represents the regularization parameter, $\mathcal{L}(y|x;\theta)$ denotes the prediction loss of using $\theta$ given the input $x$ w.r.t. response $y$, $\mathcal{D}_f$ and $\mathcal{D}_y$ refer to forget set and retain set. $y_f$ denotes the desired model response post-unlearning. The retain set represents the non-target. It indicates that, during LLM unlearning, the objective is to maintain the utility of the retain set, while simultaneously ensuring that the model avoids generating the undesired responses associated with the forget set.

### 2.2 CIRCUIT DISCOVERY AND LOGICAL CIRCUIT

In Transformer decoder-based language models, the forward pass is typically conceptualized as a **computational graph** $\mathcal{G}$, where the nodes represent **neurons**, such as output, MLPs, and query, key, and value matrices in each head and an edge $i \rightarrow j$ denotes a connection where the activation of node $i$ serves as input to node $j$. Circuit discovery seeks to identify a subgraph (circuit) $\mathcal{C} \subset \mathcal{G}$ for a target dataset that captures the task-relevant behavior (or mechanism/capability) of this dataset (Elhage et al., 2021; Conmy et al., 2023; Rai et al., 2024), with the following objective:

$$\arg\min_{\mathcal{C}} \mathbb{E}_{(x)\in\mathcal{T}}[D(p_{\mathcal{G}}(y|x)||p_{\mathcal{C}}(y|x))], \ \ s.t. \ 1 - |\mathcal{C}|/|\mathcal{G}| \geq s \tag{2}$$

where $s$ denotes the requirement of sparsity, $\mathcal{T}$ represents the target dataset, and $D$ represent the distance to quantify the difference between the two outputs from $\mathcal{G}$ and $\mathcal{C}$. Circuit discovery aims to retain the minimal $\mathcal{C}$ while faithfully reflecting the model's capability in processing the $\mathcal{T}$. The nodes and edges within this circuit are regarded as those exerting the most critical influence on the $\mathcal{T}$. We define the scope of **nodes** to include: $q$, $k$, $v$, $o$, $MLP_{up}$, and $MLP_{down}$, treating them as the smallest independent units that influence the outcome. Given that the adopted baseline model is the Zephyr-7B-beta model, we additionally include $MLP_{gate}$. Consequently, the entire computation graph, excluding inputs and outputs, consists of 32 layers, with each layer containing 7 nodes, and each node corresponds to a learnable parameter matrix. The circuit analysis examines the activation relationships between these nodes.

Increasing research has demonstrated the existence of various logical structures within circuits. Heimersheim & Nanda (2024) identified the AND and OR gates in circuits. Considering an AND gate with node $B$ as output and nodes $A_1$, $A_2$ as input, $B$ is activated only when both $A_1$ and $A_2$ are activated simultaneously. Conversely, if $B$, $A_1$, $A_2$ construct an OR gate, i.e., $B = A_1$ or $A_2$, then $B$ is activated as long as at least one of $A_1$ or $A_2$ is activated. Similar OR gates have also been observed in Conmy et al. (2023) and Wang et al. (2023). Furthermore, Chen et al. (2025) refined the classification of logical structures, proposing three distinct types: AND, OR, and ADDER[2].

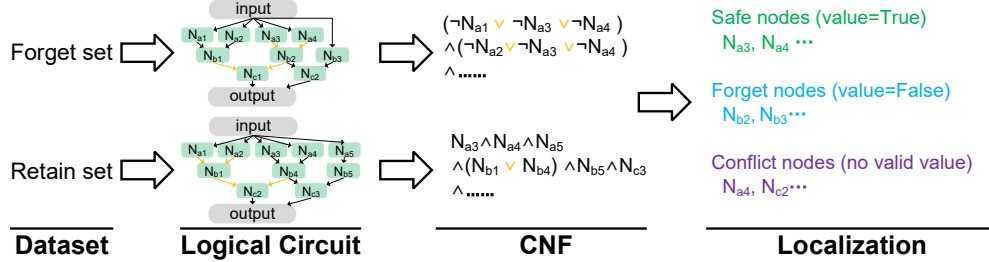

Figure 2: Overview from datasets to localization.

## 3 CONFLICT-GUIDED LOCALIZATION

In this section, we provide a three-step framework of how circuit discovery ultimately enables precise localization. An overview of our localization procedure is shown in Figure 2. Specifically,

- **Step 1** (Section 3.1). Using the circuit discovery algorithm, we separately capture the forget circuit and the retain circuit, corresponding to the forget set and the retain set, respectively. The transformation from model to circuit reveals the nodes and activation connections that are most critical for the responses to the forget and the retain set.

- **Step 2** (Section 3.2). We then transform the forget circuit and the retain circuit into CNF. The forget CNF ensures satisfiability from the perspective of forgetting, while the retain CNF ensures satisfiability from the perspective of retaining. These two CNFs provide a logical basis for assigning nodes to forget nodes, retain nodes, and conflict nodes.

- **Step 3** (Section 3.3). Finally, we jointly solve the forget and retain CNF to determine the assignment of each node. nodes with value $= 1$ influence only the retain set, whereas nodes with value $= 0$ influence only the forget set. nodes without a valid assignment (i.e., those that produce conflicts) indicate shared influence on both the forget and retain sets.

### 3.1 LOGICAL CIRCUIT DISCOVERY

We utilize the Edge-Pruning (Bhaskar et al., 2024) to build an initiative circuit and logical circuit framework (Chen et al., 2025) to further determine the logical property (AND, OR gates) of edges.

From each dataset, we extract a circuit using Eq. 2, denoted as $\mathcal{C}$, which attempts to reconstruct the functionality of the computational graph $\mathcal{G}$. Specifically, as shown in Figure 3, for the forget set, two distinct basic circuits, $\mathcal{C}_{Ns}$ and $\mathcal{C}_{Dn}$, are first obtained via noising-based intervention and denoising-based intervention, respectively. Then, based on the interaction between $\mathcal{C}_{Ns}$ and $\mathcal{C}_{Dn}$, all edges within them are classified as either AND or OR types, which are subsequently combined to form a logical circuit, designated as the **forget circuit** ($\mathcal{C}_f$) which contains all nodes and activation connections

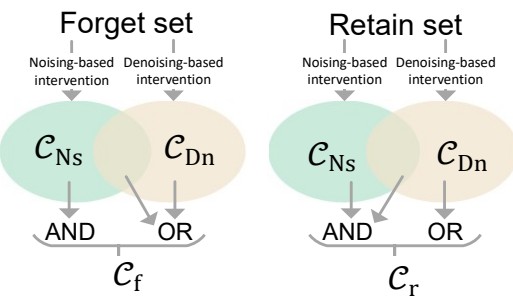

Figure 3: The processing from datasets to logical circuits.

required for the model to produce original responses that are harmful. The same methodology is applied to the retain set to derive the **retain circuit** ($\mathcal{C}_r$) which contains all nodes and activation connections necessary for the model to generate the original responses corresponding to the retain set. We elaborate on the detailed process of logical circuits in Appendix D.

---

[2]The ADDER gate, unlike binary gates such as AND and OR, represents a process where the output's effect is an accumulation of all input effects (we explain it in Appendix D). In this work, to define the CNF, we simplify the ADDER gate in the forget circuit to an OR gate, and the ADDER gate in the retain circuit to an AND gate. In Appendix C, we prove that such simplifications do not affect the unlearning functionality of the circuit.

## 3.2 CIRCUITS TO CNF

We convert the circuit into conjunctive normal form (CNF) in order to analyze the specific states of individual nodes during the unlearning task. Specifically, we apply the Tseytin transformation to convert AND and OR gates into CNF sub-expressions:

$$\begin{cases} \text{clauses} = (\neg A \vee \neg B \vee C) \wedge (A \vee \neg C) \wedge (B \vee \neg C) \text{ (if } C = A \text{ AND } B) \\ \text{clauses} = (A \vee B \vee \neg C) \wedge (\neg A \vee C) \wedge (\neg B \vee C) \text{ (if } C = A \text{ OR } B) \end{cases} \tag{3}$$

The above CNF conversion transforms $\mathcal{C}_f$ and $\mathcal{C}_r$ derived in Section 3.1 into the forget CNF $\Phi_f$ and the retain CNF $\Phi_r$. Both $\Phi_f$ and $\Phi_r$ are composed of a variable set that includes nodes (acting as A, B, or C in Eq.3) and an output (output$_f$ for $\Phi_f$ and output$_r$ for $\Phi_r$), in which all variables possess a binary value. We define the state $= 1$ (*True*) as retaining, meaning that these nodes and circuits are expected to persist in the post-unlearning model. We define state $= 0$ (*False*) as forgetting, meaning that these nodes are expected to forget certain knowledge, and the corresponding circuit is expected not to persist in the post-unlearning model. The final CNF representation is given by

$$\Phi = \Phi_f \wedge \Phi_r \wedge (\neg\text{output}_f) \wedge (\text{output}_r) \tag{4}$$

CNF $\Phi$ expects the output of $\mathcal{C}_f$ (i.e., output$_f$) to be 0 ((*False*)), indicating that the functionality of $\mathcal{C}_f$ is removed in the post-unlearning model, and it expects the output of $\mathcal{C}_r$ (i.e., output$_r$) to be 1 ((*True*)), indicating that the functionality of the $\mathcal{C}_r$ is retained in the post-unlearning model.

## 3.3 LOCALIZATION VIA CNF SOLUTION

Nodes with the same name must have identical states (1 or 0) in both $\Phi_f$ and $\Phi_r$ because they reflect whether this node should be retained or edited for unlearning. Hence, we directly solve Eq. 4 as a satisfiability problem to determine the specific state of each node.

Specifically, when $\Phi$ is **satisfiable**, all nodes with a state of 1 (*True*) indicate that they are in a preserved state, so their retention will not affect the removal of the forget circuit or destroy the retain circuit. They are unlikely to exclusively contain (or even not contain) the information to be forgotten or they have a significant causal effect on the retain circuit. Consequently, we refer to them as "**retain nodes**". In contrast, all nodes with a state of 0 (*False*) represent nodes that must be removed to ensure the forget circuit is eliminated and the retain circuit remains intact. As a result, they are likely to exclusively or necessarily contain the information to be forgotten, or they do not make a critical contribution to the retain circuit; hence, we call them "**forget nodes**". However, if $\Phi$ is found to be **unsatisfiable**, besides retain nodes and forget nodes, there will be **conflict nodes**. These are nodes that do not have a consistent value across each clause in Eq. 4. Regardless of whether these nodes have a value of 0 or 1, they cannot simultaneously satisfy the conditions of removing the forget circuit and preserving the retain circuit. Such nodes indicate that both contain necessary information to be forgotten and have an important causal effect on the retain set's response (We show the case analysis of satisfiable or unsatisfiable situation in Appendix E).

To address that, we utilize a conflict-driven clause learning SAT solver, as proposed in Zhu et al. (2025); Fleury & Heisinger (2020), to determine the satisfiability of $\Phi$. Additionally, this solver is used to find the values of all nodes under the condition of a minimum number of, or no, conflict nodes. Finally, all nodes of models can be divided into one of the following types:

**Safe nodes**: These include retain nodes (state=1) and nodes that do not appear in $\Phi$. Safe nodes are irrelevant to forget set, and do not require any editing.

**Forget nodes**: These are nodes with a value of 0 (False) in $\Phi$. Forget nodes do not impact the response of the retain set and merely have harmful information which needs to be removed to enhance the forgetting efficacy.

**Conflict nodes**: These are nodes for which no valid assignment exists that satisfies $\Phi$. On one hand, conflict nodes must be modified to remove harmful information; on the other hand, modifying these nodes can lead to a decrease in the retain set's performance.

## 4 UNLEARNING VIA CONFLICT-GUIDED LOCALIZATION

Based on the forget and conflict nodes obtained from localization, we adopt a **two-stage fine-tuning approach** for unlearning. First, we generate a **forget mask** ($\mathcal{M}_\mathrm{f}$) and **conflict mask** ($\mathcal{M}_\mathrm{c}$) with a parameter scale equal to the language model[3]. In $\mathcal{M}_\mathrm{f}$, all elements corresponding to the forget node are set to 1, while all others are set to 0. Similarly, in $\mathcal{M}_\mathrm{c}$, all elements corresponding to the conflict node are set to 1, while all others are set to 0.

In the **first stage**, we begin by fine-tuning the **forget nodes**. Since forget nodes do not significantly influence the response of the retain set, we only use the forget loss to constrain the fine-tuning. Thus, the unlearning problem in this stage is defined as:

$$\min_{\theta_\mathrm{f}} \mathbb{E}_{(x,y_f)\in\mathcal{D}_f}[\mathcal{L}(y_f|x; \mathcal{M}_\mathrm{f} \odot \theta_\mathrm{f} + (1 - \mathcal{M}_\mathrm{f}) \odot \theta_\mathrm{o})] \tag{5}$$

where $\theta_\mathrm{f}$ represents the parameters of forget nodes and $\theta_\mathrm{o}$ represents parameters of other nodes.

In the **second stage**, we further fine-tune the **conflict nodes**. Conflict nodes have a significant causal effect on the responses of both the forget set and the retain set, so fine-tuning requires constraints from both the forget and retain loss. Therefore, the unlearning problem is formulated as:

$$\min_{\theta_\mathrm{c}} \mathbb{E}_{(x,y_f)\in\mathcal{D}_f}[\mathcal{L}(y_f|x; \mathcal{M}_\mathrm{f} \odot \theta_\mathrm{c} + (1 - \mathcal{M}_\mathrm{f}) \odot \theta_\mathrm{o})] + \lambda \mathbb{E}_{(x,y)\in\mathcal{D}_r}[\mathcal{L}(y|x; \mathcal{M}_\mathrm{f} \odot \theta_\mathrm{c} + (1 - \mathcal{M}_\mathrm{f}) \odot \theta_\mathrm{o})] \tag{6}$$

where $\theta_\mathrm{c}$ represents the parameters of forget nodes. The retain loss $\mathcal{L}(y|x; \mathcal{M}_\mathrm{f} \odot \theta_\mathrm{c} + (1 - \mathcal{M}_\mathrm{f}) \odot \theta_\mathrm{o})$ typically mirrors the training loss over the retain set. However, for the forget loss, there are various types of implementations, such as GA (Liu et al., 2022), NPO (Zhang et al., 2024), and PO (Maini et al., 2024). In this paper, we adopt PO for main results and show the ablation study about GA and NPO.

## 5 EXPERIMENT SETUPS

### 5.1 UNLEARNING TASKS, DATASETS, MODELS, AND BASELINES

We conduct experiments on **three** LLM unlearning tasks[4], where each task is assigned **four** retain datasets. These three tasks are: **WMDP Cyber** (Li et al., 2024), which focuses on malicious use prevention of LLMs in developing cyberattacks; **WMDP Bio** (Li et al., 2024), which assesses the capability to prevent the hazardous knowledge in biosecurity; **PKU-SafeRLHF** (Ji et al., 2023), which aims to prevent the toxic content in response to inappropriate prompts from SafeRLHF. For the retain set, to better observe specific circuits, we selected four datasets with distinct tasks/capabilities rather than original general corpus: **Winogrande** (ai2, 2019), which involves the task to infer the correct referent of a pronoun from semantics; **SST-2** (Socher et al., 2013), which includes the task of inferring sentiment categories from a given text; **RTE** (Recognizing Textual Entailment) (Dagan et al., 2022), which involves a model determining the entailment relationship between two texts; and **Bool** (Suzgun et al., 2023), which includes the task of performing logical operations. Model-wise, we follow existing practices and use the Zephyr-7B-beta model (Tunstall et al., 2024) for WMDP Cyber and WMDP Bio, and LLaMA2-7B (Touvron et al., 2023) for PKU-SafeRLHF.

CLUE is a framework that performs both localization and fine-tuning, so we select two categories of baselines for comparison. The first category includes localization methods for LLM unlearning, such as **WAGLE** (Jia et al., 2024), **DEPN** (Wu et al., 2023), **MEMIT** (Patil et al., 2023), and **PCGU** (Yu et al., 2023). The second category comprises fine-tuning methods for LLM unlearning, including **GA** (Yao et al., 2024), **NPO** (Zhang et al., 2024), and **PO** (Maini et al., 2024) (Further details ain Appendix F).

---

[3]Taking the Zephyr-7B-beta model as an example, each layer contains seven parameter matrices: $q$, $k$, $v$, $o$, $MLP_{gate}$, $MLP_{up}$, and $MLP_{down}$. There are 32 layers in total, so there are 224 parameter matrices in the mask. The size of each parameter matrix is determined by the actual dimensions of the Zephyr-7B-beta model, such as the $q$ matrix, which has dimensions $[4096 \times 4096]$.

[4]We do not evaluate on the TOFU (Maini et al., 2024) and Who's Harry Potter (Eldan & Russinovich, 2023) datasets. This is because these datasets require a self-fine-tuned model as the baseline, and such fine-tuning affects the circuits of non-target tasks, which leads to a lack of credibility in the results.

## 5.2 Training and Evaluation Setup

To obtain LLMs post-unlearning, we adopt PO as forget loss which performs better than GA and NPO (Jia et al., 2024). All fine-tuning processes are conducted over 6 epochs, with 1 epoch for forget nodes and the rest 5 epochs for conflict nodes. The learning rate is grid-searched at $1 \times 10^{-5}$ for each dataset. The parameter $\lambda = 1$, and we adopted AdamW (Loshchilov & Hutter, 2017) as the optimizer. All experiments were conducted on 16 NVIDIA RTX A100 GPUs.

We evaluate the performance of unlearned LLMs from **forgetting efficacy** and **retaining utility**. Forgetting efficacy adopts accuracy of LLMs post-unlearning on the forget set as the main metric. For aligned tendency, we use 1-accuracy to measure forgetting efficacy. Thus, a higher 1-accuracy indicates better unlearning. Moreover, we also provide the efficacy results about other prevalent metrics, such as Membership inference attack (MIA) and Rouge-L, with detailed results shown in Appendix H. Next, we measure retaining utility with **accuracy** in **both retain set and other non-target tasks**. Specifically, for each retain dataset (one of Winogrande, SST-2, RTE, and Bool), we first measure the accuracy on its corresponding test set (retain utility). Subsequently, we measure the average accuracy on a series of unrelated tasks (general utility). These unrelated tasks were evaluated using the **Language Model Evaluation Harness** toolkit (Gao et al., 2021) and include: **ARC-Challenge** (Chollet, 2019), **ARC-Easy** (Chollet, 2019), **BoolQ** (Clark et al., 2019), **HellaSwag** (Zellers et al., 2019), **OpenBookQA** (Mihaylov et al., 2018), **Piqa** (Bisk et al., 2020), and **TruthfulQA** (Lin et al., 2021). Details about evaluation datasets are shown in Appendix F.

## 6 Experiment Results

### 6.1 Can CLUE Improve LLM Unlearning Through Localization?

Table 1: Performance overview of LLM unlearning. "Unlearned Parameter" refers to the percentage of parameters modified, calculated by averaging the percentage of changes in each parameter matrix. "FE" (Forget efficacy) is measured as 1-accuracy and "RU" (Retain utility) is measured as accuracy on the test set of the retain set. "GU" (General utility) is average accuracy on a series of non-target tasks, and specific results can be found in Appendix G.

| Method | Unlearned Parameter | Retain Set | | | | | | | | | | | |
| | | Winogrande | | | SST-2 | | | RTE | | | Bool | | |
| | | FE↑ | RU↑ | GU↑ | FE↑ | RU↑ | GU↑ | FE↑ | RU↑ | GU↑ | FE↑ | RU↑ | GU↑ |
| **WMDP Cyber** | | | | | | | | | | | | | |
| Origin | - | 0.445 | 0.729 | 0.624 | 0.445 | 0.727 | 0.624 | 0.445 | 0.703 | 0.624 | 0.445 | 0.555 | 0.624 |
| GA | 100% | 0.658 | 0.254 | 0.332 | 0.665 | 0.151 | 0.312 | 0.647 | 0.067 | 0.367 | 0.652 | 0.124 | 0.325 |
| NPO | 100% | 0.663 | 0.342 | 0.329 | 0.654 | 0.247 | 0.336 | 0.697 | 0.264 | 0.339 | 0.654 | 0.237 | 0.365 |
| PO | 100% | 0.685 | 0.567 | 0.368 | 0.672 | 0.632 | 0.352 | 0.685 | 0.471 | 0.374 | 0.682 | 0.317 | 0.384 |
| MEMIT | 76.27% | 0.669 | 0.662 | 0.384 | 0.673 | 0.567 | 0.359 | 0.675 | 0.524 | 0.386 | 0.690 | 0.335 | 0.386 |
| PCGU | 86.77% | 0.672 | 0.654 | 0.382 | 0.672 | 0.692 | 0.362 | 0.669 | 0.546 | 0.375 | 0.695 | 0.314 | 0.376 |
| DEPN | 78.82% | 0.695 | 0.817 | 0.431 | 0.702 | 0.731 | 0.379 | 0.712 | 0.457 | 0.416 | 0.715 | 0.429 | 0.396 |
| WAGLE | 90.01% | **0.702** | 0.86 | 0.442 | 0.708 | 0.771 | 0.384 | 0.685 | 0.498 | 0.413 | 0.721 | 0.434 | 0.387 |
| CLUE | **58.16%** | 0.697 | **0.992** | **0.458** | **0.733** | **0.91** | **0.388** | **0.744** | **0.786** | **0.436** | **0.724** | **0.505** | **0.434** |
| **WMDP Bio** | | | | | | | | | | | | | |
| Origin | - | 0.355 | 0.729 | 0.624 | 0.355 | 0.727 | 0.624 | 0.355 | 0.703 | 0.624 | 0.355 | 0.555 | 0.624 |
| GA | 100% | 0.564 | 0.064 | 0.375 | 0.675 | 0.124 | 0.379 | 0.564 | 0.125 | 0.385 | 0.568 | 0.214 | 0.354 |
| NPO | 100% | 0.571 | 0.241 | 0.372 | 0.671 | 0.234 | 0.385 | 0.574 | 0.269 | 0.374 | 0.572 | 0.315 | 0.384 |
| PO | 100% | 0.605 | 0.421 | 0.385 | 0.685 | 0.446 | 0.382 | **0.589** | 0.321 | 0.385 | 0.585 | 0.385 | 0.381 |
| MEMIT | 74.29% | 0.591 | 0.672 | 0.429 | 0.695 | 0.619 | 0.399 | 0.547 | 0.395 | 0.421 | 0.605 | 0.421 | 0.395 |
| PCGU | 85.12% | 0.601 | 0.662 | 0.415 | 0.684 | 0.627 | 0.402 | 0.539 | 0.402 | 0.419 | 0.596 | 0.413 | 0.396 |
| DEPN | 77.24% | 0.605 | 0.739 | 0.469 | 0.701 | 0.761 | 0.424 | 0.546 | 0.443 | 0.471 | 0.599 | 0.429 | 0.441 |
| WAGLE | 90.02% | 0.599 | 0.885 | 0.480 | 0.698 | 0.785 | 0.426 | 0.549 | 0.466 | 0.472 | 0.601 | 0.412 | 0.441 |
| CLUE | **56.19%** | **0.617** | **0.995** | **0.499** | **0.713** | **0.893** | **0.457** | 0.586 | **0.528** | **0.491** | **0.612** | **0.501** | **0.456** |
| **PKU-SafeRLHF** | | | | | | | | | | | | | |
| origin | - | 0.294 | 0.841 | 0.664 | 0.294 | 0.764 | 0.664 | 0.294 | 0.795 | 0.664 | 0.294 | 0.514 | 0.664 |
| GA | 100% | 0.615 | 0.124 | 0.394 | 0.605 | 0.095 | 0.385 | 0.625 | 0.147 | 0.360 | 0.601 | 0.054 | 0.385 |
| NPO | 100% | 0.605 | 0.195 | 0.385 | 0.612 | 0.154 | 0.395 | 0.614 | 0.196 | 0.327 | 0.616 | 0.214 | 0.396 |
| PO | 100% | 0.625 | 0.361 | 0.395 | 0.623 | 0.225 | 0.396 | 0.623 | 0.314 | 0.395 | 0.625 | 0.387 | 0.402 |
| MEMIT | 77.62% | 0.645 | 0.545 | 0.402 | 0.649 | 0.395 | 0.402 | 0.625 | 0.436 | 0.409 | 0.647 | 0.359 | 0.412 |
| PCGU | 86.29% | 0.639 | 0.625 | 0.400 | 0.633 | 0.399 | 0.404 | 0.639 | 0.397 | 0.417 | 0.639 | 0.402 | 0.406 |
| DEPN | 74.36% | 0.661 | 0.794 | 0.412 | 0.657 | 0.741 | 0.429 | 0.634 | 0.421 | 0.415 | 0.642 | 0.422 | 0.429 |
| WAGLE | 90.01% | 0.655 | 0.751 | 0.429 | 0.663 | 0.761 | 0.434 | 0.641 | 0.496 | 0.421 | 0.635 | 0.422 | 0.411 |
| CLUE | **54.88%** | **0.724** | **0.956** | **0.462** | **0.681** | **0.883** | **0.455** | **0.656** | **0.682** | **0.429** | **0.659** | **0.536** | **0.438** |

To investigate the performance of CLUE compared to existing methods, we test the performance of existing methods and CLUE on four different retain sets across the WMDP Cyber, WMDP Bio, and PKU-SafeRLHF datasets. Table 1 reports the results of these performances. All experiments were repeated five times, and the standard deviation was omitted as it was consistently less than 0.01.

It is clear that localization-based methods (MEMIT, PCGU, DEPN, WAGLE, CLUE) significantly outperform finetuning-based methods (GA, NPO, PO) in terms of both retain utility and general utility. We attribute this to the fact that localization can, to some extent, filter out important nodes, thereby preventing the capabilities for non-target tasks from being affected. Furthermore, CLUE generally outperforms existing methods in forget efficacy, retain and general utility. The advantage in forget efficacy and utility comes from our more precise localization of forget nodes and conflict nodes, which prevents a large number of irrelevant nodes from being fine-tuned.

To further validate the roles of the forget mask ($\mathcal{M}_{\mathrm{f}}$) and the conflict mask ($\mathcal{M}_{\mathrm{c}}$), we conduct ablation studies on the WMDP Cyber dataset using CLUE. We choose SST-2 as the retain set. The specific ablation measures are: "-$\mathcal{M}_{\mathrm{f}}$": The fine-tuning process for the forget mask is removed, and only the conflict mask is used for fine-tuning. "-$\mathcal{M}_{\mathrm{c}}$": The conflict mask is replaced with a full-true mask matrix (in this mask, all values=1). "$\mathcal{M}_{\mathrm{c}}$ to $\mathcal{M}_{\mathrm{f}}$": Fine-tuning is performed first on the conflict mask and then on the forget mask. Additionally, we investigat the impact of different fine-tuning methods on CLUE. The default fine-tuning is "PO+PO", where both the first and second stages use PO for fine-tuning. We then test various combinations where each stage is replaced with "GA" or "NPO".

Table 2 shows that when the forget mask is removed, the forget efficacy decreases the most. This supports the importance of forget nodes for information forgetting. Similarly, replacing the conflict mask also leads to the largest drop in utility, which indicates that the conflict mask is effective at preventing irrelevant nodes from being fine-tuned. Moreover, the GA method results in a significant performance decrease, especially in utility. This is consistent with the conclusion in Zhang et al. (2024) that GA leads to catastrophic forgetting by causing large-scale modifications to node parameters.

Additionally, in Appendix H, we present the results for the experiments in Table 1 on the MIA and Rouge-L metrics, which also demonstrate that our method consistently outperforms existing approaches. In Appendix I, we show the performance of LLM unlearning varying different forget ratios, which indicates that the conflict-guided localization is beneficial for the unlearning task across different forget ratios.

Table 2: Ablation with WMDP Cyber as forget set and SST-2 as retain set.

| Method | forget efficacy | retain utility | general utility |
|---|---|---|---|
| CLUE | 0.733 | 0.91 | 0.388 |
| -$\mathcal{M}_{\mathrm{f}}$ | ↓**0.045** | ↓0.007 | ↓0.011 |
| -$\mathcal{M}_{\mathrm{c}}$ | ↓0.005 | ↓**0.264** | ↓**0.053** |
| $\mathcal{M}_{\mathrm{c}}$ to $\mathcal{M}_{\mathrm{f}}$ | ↓0.024 | ↓0.192 | ↓0.019 |
| GA+GA | ↓**0.021** | ↓**0.529** | ↓**0.026** |
| GA+PO | ↓0.012 | ↓0.067 | ↓0.003 |
| PO+GA | ↓0.005 | ↓0.191 | ↓0.012 |
| NPO+NPO | ↓0.009 | ↓0.093 | ↓0.007 |
| NPO+PO | ↓0.002 | ↓0.041 | ↓0.006 |
| PO+NPO | ↓0.004 | ↓0.081 | ↓0.008 |

## 6.2 How CLUE performs when a General Corpus serves as the Retain Set?

Another question worth exploring is how CLUE performs where the retain set is a general corpus. To investigate this, we conduct two types of experiments on WMDP Cyber: **1.** CLUE paired with the **MMLU dataset** (Hendrycks et al., 2020) (a general corpus as the default retain set for the WMDP) as the retain set. **2.** CLUE paired with **multiple datasets from specific tasks** as the retain set.

We select 12 specific tasks in total (for option 2): Winogrande, SST-2, RTE, Bool, Induction (Conmy et al., 2023), IOI (Wang et al., 2023), Gender Bias (Vig et al., 2020), Docstring (Heimersheim & Janiak, 2023), Great Than (Hanna et al., 2023), SA (Yu et al., 2024), arithmetic (Ghazal et al., 2013), Reverse (Conmy et al., 2023). We then evaluate CLUE's performance when 1 to 10 of these specific tasks are used as the retain set. Each time, we randomly sample 20 times from the 12 specific tasks based on the predetermined number of tasks. For comparison, we also provide the performance of WAGLE when MMLU is used as the retain set. Figure 4 reports both experiments. The x-axis indicates the number of retained tasks we select from the 12 specific tasks. As its number increases, we plot the forget efficacy, utility of MMLU and general utility with blue lines. While the dotted lines evaluates these three metrics when MMLU serves as the retain set.

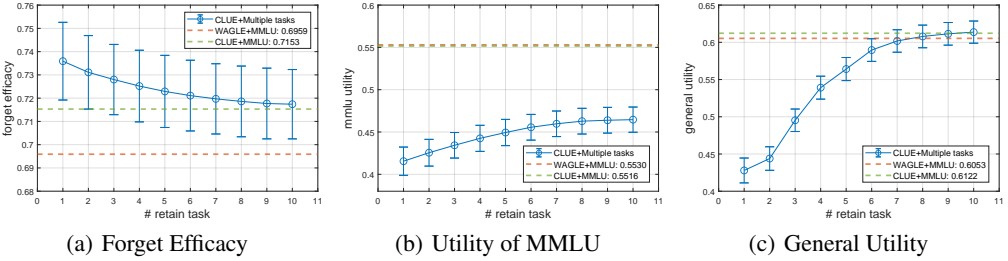

(a) Forget Efficacy  (b) Utility of MMLU  (c) General Utility

Figure 4: Performance of CLUE when retain set is MMLU dataset and multiple specific tasks.

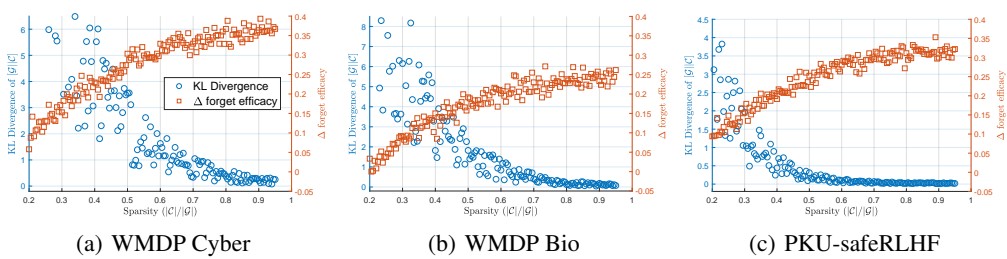

(a) WMDP Cyber  (b) WMDP Bio  (c) PKU-safeRLHF

Figure 5: Circuit Sparsity vs. Circuit Faithfulness and Forget Efficacy.

Upon examining Figure 4, we can see that even when using a general corpus like MMLU as the retain set, CLUE still demonstrates a higher forget efficacy and nearly equal utility than WAGLE. When this is combined with the conclusions from Section 6.1, we can infer that although MMLU may not provide sufficiently specific task circuits (because it is a multi-task dataset), it can still identify enough "forget nodes" via the forget set. Figure 4 (a) shows that when a specific task is used as the retain set, the forget efficacy is higher than with MMLU, but it decreases as the number of specific tasks increases. This makes sense: more specific tasks lead to more "conflict nodes," which makes unlearning critical information more difficult. Figure 4 (b) and (c) both indicate that as the number of specific tasks increases, utility shows an upward trend. Because the more specific tasks there are, the more skills or capabilities need to be retained. Although the utility on MMLU is not as high as when MMLU provides direct supervision, at 7 number of specific tasks, the general utility can finally reach and surpass the performance when MMLU is the retain set. This also suggests that when a sufficient number of specific tasks are used as the retain set, the model's utility is better preserved.

## 6.3 UNLEARNING PERFORMANCE VS. CIRCUIT SPARSITY AND FAITHFULNESS

In this section, we investigate the relationship between unlearning performance and circuit quality, especially circuit **sparsity** and **faithfulness**. Sparsity is calculated by $\frac{C}{G}$, where $\frac{C}{G}$ closer to 0 indicates fewer edges in the circuit, so more sparse. Faithfulness refers to the discrepancy between the circuit output and the computational graph output. We quantify this discrepancy using the Kullback–Leibler (KL) divergence of the output logits, where a smaller KL divergence indicates that the circuit's output is closer to the original model's output, thus demonstrating higher faithfulness.

Furthermore, a trade-off inherently exists: a more sparse circuit generally leads to lower faithfulness. Therefore, we analyze the forget efficacy of different circuits by controlling sparsity from 0.2 to 0.95. We use SST-2 as the retain set and evaluate the $\Delta$ forget efficacy (i.e., forget efficacy − original efficacy) on WMDP Cyber, WMDP Bio, and PKU-SafeRLHF datasets. Figure 5 confirms that as the sparsity decreases, the KL divergence between the circuit and the computational graph gradually decreases, while the $\Delta$ forget efficacy progressively increases. It shows that the denser the circuit, the smaller the functional gap with the computational graph, and the higher the forget efficacy. Empirically and from the results, the performance of unlearning generally reaches the optimal range when sparsity attains a level of 0.7. Therefore, all remaining experiments in this paper utilize circuits with a sparsity of 0.7.

### 6.4 HOW DOES NODE LOCALIZATION CHANGE AFTER UNLEARNING?

Table 3: The number of forget nodes and conflict nodes before and after the unlearning.

| Forget Set | Status of Unlearning | Winogrande forget node(%) | Winogrande conflict node(%) | SST-2 forget node(%) | SST-2 conflict node(%) | RTE forget node(%) | RTE conflict node(%) | Bool forget node(%) | Bool conflict node(%) |
|---|---|---|---|---|---|---|---|---|---|
| WMDP Cyber | before | $15.66_{\pm0.24}$ | $32.29_{\pm1.72}$ | $3.74_{\pm0.16}$ | $44.21_{\pm1.38}$ | $4.98_{\pm0.05}$ | $43.09_{\pm1.71}$ | $3.90_{\pm0.07}$ | $44.09_{\pm1.55}$ |
| | after | $19.47_{\pm0.27}$ | $29.47_{\pm1.31}$ | $8.61_{\pm0.17}$ | $28.01_{\pm1.56}$ | $11.94_{\pm0.39}$ | $34.71_{\pm1.39}$ | $18.94_{\pm0.74}$ | $16.33_{\pm1.72}$ |
| WMDP Bio | before | $15.62_{\pm0.23}$ | $32.31_{\pm1.71}$ | $3.85_{\pm0.15}$ | $44.08_{\pm1.41}$ | $5.02_{\pm0.05}$ | $43.06_{\pm1.71}$ | $3.95_{\pm0.07}$ | $44.12_{\pm1.52}$ |
| | after | $18.49_{\pm0.22}$ | $25.94_{\pm1.41}$ | $6.57_{\pm0.13}$ | $31.09_{\pm1.42}$ | $9.56_{\pm0.29}$ | $35.19_{\pm1.35}$ | $16.91_{\pm0.53}$ | $20.55_{\pm1.67}$ |
| PKU-SafeRLHF | before | $13.46_{\pm0.52}$ | $30.63_{\pm1.05}$ | $23.03_{\pm0.66}$ | $21.06_{\pm0.53}$ | $0.73_{\pm0.00}$ | $43.36_{\pm1.39}$ | $0.42_{\pm0.00}$ | $39.85_{\pm1.37}$ |
| | after | $18.77_{\pm0.67}$ | $22.57_{\pm1.29}$ | $28.67_{\pm0.79}$ | $15.49_{\pm0.44}$ | $5.69_{\pm0.09}$ | $31.24_{\pm1.24}$ | $3.95_{\pm0.03}$ | $26.39_{\pm1.11}$ |

In this section, we investigate whether **forget nodes** and **conflict nodes** change post-unlearning. Table 3 presents the percentages of forget and conflict nodes for different pairings of forget and retain sets, and it tracks their post-unlearning results. We observe that the proportion of forget nodes significantly increases after unlearning, while the proportion of conflict nodes decreases. This suggests that the conflict nodes, which is supervised by both a forget loss and a retain loss, shift the forget and retain circuit to less overlapping locations. This implies that CLUE's learning on conflict nodes is effective at decoupling the forget circuit from those of other capabilities or mechanisms.

Finally, we explore the specific distribution of the forget and conflict nodes to further analyze the response of CLUE to different nodes. Detailed results can be found in Appendix J. In simple terms, nearly all MLPs are conflict nodes, which aligns with the finding that MLPs typically store a large amount of information. Furthermore, compared to other methods, MEMIT appears to lack the forget nodes, while WAGLE does not differentiate between forget nodes and conflict nodes.

## 7 CONCLUSION AND LIMITATION

In this paper, we introduce CLUE, a localization framework that uses circuit discovery to identify the circuits for the forget and retain sets and converts them into a CNF. By employing a satisfiability solver, we determine the role of each node in the unlearning task, classifying them as forget nodes, retain nodes, or conflict nodes. We then provide targeted fine-tuning strategies for each type of nodes. Compared to other localization methods, CLUE offers more precise node localization and significantly outperforms existing methods in both forget efficacy and retain utility.

However, CLUE still has some limitations that can be explored further. First, the circuit is static and cannot dynamically reflect changes in key nodes during the fine-tuning process. Therefore, exploring the dynamics of circuits during parameter fine-tuning is a direction for future work. Additionally, when dealing with multiple retain sets, although CLUE can identify which nodes are associated with specific combinations of retain sets, the fine-tuning stage cannot provide targeted fine-tuning solutions for every possible conflict combination. Consequently, developing editing methods other than fine-tuning is another key focus of our future research. Moreover, we must acknowledge that there may exist potential forget datasets from which a clear circuit cannot be extracted (possibly due to multi-domain mixing or excessive bias). Fortunately, this issue has not been observed in current mainstream forget datasets; however, this remains a potential risk. Lastly, the inherent scalability problems of circuit discovery also pose a significant challenge, making CLUE difficult to apply to very large-scale language models. Nevertheless, current research (Syed et al., 2024; Lieberum et al., 2023) is attempting to establish the feasibility of circuit discovery methods on these larger models. Thus, the potential to resolve the scalability issues for CLUE remains a promising area of investigation.

## ACKNOWLEDGMENTS

This research/project is supported by the National Research Foundation, Singapore under its National Large Language Models Funding Initiative, (AISG Award No: AISG-NMLP-2024-005), the NTU Start-Up Grant (#023284-00001), Singapore, and the MOE AcRF Tier 1 Seed Grant (RS37/24, #025041-00001), Singapore.

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

## A    REPRODUCIBILITY STATEMENT

Our code is publicly available in an anonymized repository linked in the abstract, which contains the complete implementation of CLUE and related instructions (see README). Furthermore, each theoretical step presented in the paper is supported by citations to relevant research (Chen et al., 2025; Zhu et al., 2025). For clarification, we provide illustrative examples in Appendices C and D. All datasets required for our experiments are available on Hugging Face and are described in detail within the manuscript (see Section 5).

## B    THE USE OF LARGE LANGUAGE MODELS

In the preparation of this paper, we utilized a large language model (LLM) as an assistive tool toenhance the quality of our writing and presentation. The LLM's role was strictly confined to refining the manuscript's writing and formatting, without generating any core scientific content or data.

## C    EXAMPLE FROM CIRCUIT TO CNF

Following the notion of logical circuits introduced in (Chen et al., 2025), we construct a toy circuit as illustrated in Figure 6. The semantics of the three gates are defined as follows:

**AND gate**. The receiver node is activated if and only if the activation states of all sender nodes equal 1; otherwise, the receiver node remains inactive.

**OR gate**. The receiver node is inactive if and only if the activation states of all sender nodes equal 0; otherwise, the receiver node is activated.

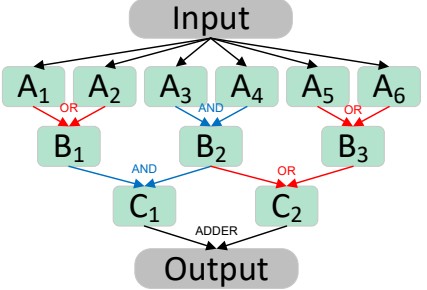

Figure 6: The toy circuit with AND, OR, and ADDER gate.

**ADDER gate**. Unlike the previous two gates, the receiver node in this case admits multiple intermediate activation states rather than a binary activated/inactivated outcome. Each intermediate state corresponds to the contribution of a sender node, with all sender nodes treated as equally weighted and independent. For example, if only one sender node has activation state equal to 1, then the receiver node has activation state 1; if two sender nodes are active, then the receiver node has activation state 2, and so on. In general, the logical relation is expressed as receiver node = sender node$_1$ + sender node$_2$ + $\cdots$.

We use an activation state of 1 to indicate that a given node should be retained and state of 0 to indicate that it should be unlearned. To preserve the maximal capability of the retain set, the gates are governed by the following rules:

- AND gate: all sender nodes must be retained.

- OR gate: at least one sender node must be retained.

- ADDER gate: all sender nodes must be retained; otherwise, the receiver node fails to reach its optimal activation value, thereby impairing the model's capability on the retain set.

Consequently, we define both the AND gate and the ADDER gate as conjunctions (denoted by the symbol $\wedge$), while the OR gate is defined as a disjunction (denoted by the symbol $\vee$). In this definition, the ADDER gate only has two states (0/1). For the toy circuit in Figure 6, this leads to the following conjunctive normal form (CNF):

$$\text{output} = C_1 \wedge C_2 \wedge B_1 \wedge B_2 \wedge (B_2 \vee B_3) \wedge (A_1 \vee A_2) \wedge A_3 \wedge A_4 \tag{7}$$

Analogously, for the forget set, we represent the state of each gate using a negation operator ($\neg$), since the capability of the forget set is ideally satisfied when all nodes in the circuit have activation

state equal to 0. In this setting, the AND gate is expressed as a disjunction. The reason is that, to unlearn the capability associated with the forget set, it suffices to unlearn any one of the incoming edges of an AND gate. For example, in Figure 6, we have $\neg C_1 = \neg B_1 \vee \neg B_2$ .

By contrast, both the OR gate and the ADDER gate are expressed as conjunctions. This is because all of their sender nodes must be unlearned to ensure that the capability of the entire gate is forgotten. For instance, $\neg B_3 = \neg A_5 \wedge \neg A_6$ .

Therefore, if Figure 6 corresponds to the forget set, the resulting conjunctive normal form (CNF) can be written as

$$
\begin{aligned}
\text{output} = & \neg C_1 \wedge \neg C_2 \wedge (\neg B_1 \vee \neg B_2) \wedge \neg B_2 \neg B_3 \wedge \\
& (\neg A_1 \vee \neg A_3 \vee \neg A_4) \wedge (\neg A_2 \vee \neg A_3 \vee \neg A_4) \wedge (\neg A_3 \vee \neg A_4) \wedge \neg A_5 \wedge \neg A_6
\end{aligned}
\tag{8}
$$

Evidently, the CNF corresponding to the retain set is always satisfiable when all sender nodes satisfy state=1, and the CNF corresponding to the forget set is always satisfiable when all sender nodes satisfy state=0.

The above analysis proves that in the forget circuit, the propositional logic of the ADDER gate is the same as that of the OR gate, while in the retain circuit, the propositional logic of the ADDER gate is the same as that of the AND gate. Therefore, in the actual implementation, we convert the ADDER gate in the forget circuit into an OR gate and the ADDER gate in the retain circuit into an AND gate. Then, we use the logically complete Tseytin transformation to convert them into CNF.

## D    DETAILS OF LOGICAL CIRCUIT FRAMEWORK

At first, we systematically introduce three fundamental circuit logic types: the **AND** gate, **OR** gate, and **ADDER** gate (Chen et al., 2025).

**Definition 1.** *We assume a common paradigm in which a receiver node $B$, which is connected by more than 1 sender node $A_1, A_2, \ldots$. For any edge $A_i \rightarrow B$, we use binary values '0' and '1' to represent the activation state of a node. Specifically, $A_i = 0$ indicates that node $A_i$ is removed, ablated, or deactivated, whereas $A_i = 1$ indicates that node $A_i$ is retained and active. When the sender nodes are ablated, the effect of node $B$ on the output exhibits three distinct patterns, which are as follows:*

**AND***: All sender nodes satisfy an AND logical relationship with the receiver node, i.e., $B = A_1 \wedge A_2 \wedge \ldots$. In this case, node $B$ exerts a significant effect on the output only if all of its sender nodes are retained. If even a single sender node is ablated, the effect of $B$ on the output is nearly eliminated.*

**OR** *gate: All sender nodes satisfy an OR logical relationship with the receiver node, i.e., $B = A_1 \vee A_2 \vee \ldots$. In this case, node $B$ always exerts a significant effect on the output if one or more of its sender nodes are retained. Only if all sender nodes are ablated, the effect of $B$ on the output is nearly eliminated.*

**ADDER** *gate: all sender nodes satisfy an ADDER logical relationship with the receiver node, i.e., $B = A_1 + A_2 + \ldots$. In this case, node $B$ exhibits its maximal effect on the output only when all of its sender nodes are retained. If any single sender node is ablated, the effect of $B$ on the output is substantially diminished; when all sender nodes are ablated, $B$'s effect on the output is reduced to zero. Accordingly, we define the state of $B$ as taking values 0,1,2,..., where the total number of distinct states equals the number of sender nodes.*

Theoretical analyses support the view that noising-based intervention is capable of recovering a complete AND gate but fails to recover a complete OR gate, whereas denoising-based intervention demonstrates the opposite pattern (Heimersheim & Nanda, 2024). This asymmetry is straightforward to interpret. The noising-based intervention procedure corresponds to the transition from a clean activation state (state = 1) to a corrupted activation state (state = 0). Since all gates can be regarded as being initialized with activation states equal to 1, any transition to state = 0 induces a significant change in the effect of AND and ADDER gates on the output. Consequently, noising-based intervention can reliably identify AND and ADDER gates.

The **denoising-based intervention** first performs the corrupted run in the computational graph, and then replaces the corrupted activations with the clean activations. Those activations that lead to significant changes in the output ($\tilde{y}$) consist of the circuits. denoising-based intervention thus has the following objective:

$$\arg\min_{\mathcal{C}} \mathbb{E}_{(x,\tilde{x}) \in \mathcal{T}}[D(p_{\mathcal{G}}(\tilde{y}|\tilde{x})||p_{\mathcal{C}}(\tilde{y}|\tilde{x},x))], \quad s.t.\ 1 - |\mathcal{C}|/|\mathcal{G}| \geq s \tag{9}$$

Conversely, the denoising-based intervention procedure corresponds to initialization with activation states equal to $0$. In this case, any transition to state $= 1$ produces a significant change in the effect of OR and ADDER gates on the output.

Therefore, we denote the circuit constructed under the noising-based intervention strategy as $\mathcal{C}_{\text{Ns}}$, and the one constructed under the denoising-based intervention strategy as $\mathcal{C}_{\text{Dn}}$. Based on the above set-theoretic relationships between $\mathcal{C}_{\text{Ns}}$ and $\mathcal{C}_{\text{Dn}}$, we extract subsets of edges corresponding to AND, OR, and ADDER gates as follows:

- AND gate ($\mathcal{C}_{\text{AND}}$): edges that are present in $\mathcal{C}_{\text{Ns}}$ but absent from $\mathcal{C}_{\text{Dn}}$.
- OR gate ($\mathcal{C}_{\text{OR}}$): edges that are present in $\mathcal{C}_{\text{Dn}}$ but absent from $\mathcal{C}_{\text{Ns}}$.
- ADDER gate ($\mathcal{C}_{\text{ADDER}}$): edges that are shared between $\mathcal{C}_{\text{Ns}}$ and $\mathcal{C}_{\text{Dn}}$.

Therefore, we propose a combined **Ns+Dn** approach to recover logically complete gates. This method is compatible with a wide range of circuit discovery algorithms, introduces minimal additional computational overhead, and enables clear and effective separation of the three types of logic gates. Ns+Dn has the following objective:

$$\arg\min_{\mathcal{C}} \mathbb{E}_{(x,\tilde{x}) \in \mathcal{T}}[D(p_{\mathcal{G}}(y|x)||p_{\mathcal{C}}(y|x,\tilde{x})) + D(p_{\mathcal{G}}(\tilde{y}|\tilde{x})||p_{\mathcal{C}}(\tilde{y}|\tilde{x},x))], \quad s.t.\ 1 - |\mathcal{C}|/|\mathcal{G}| \geq s \tag{10}$$

Finally, we simplify the ADDER gate in the forget circuit to an OR gate, and the ADDER gate in the retain circuit to an AND gate, as shown in Appendix C.

# E  CASE ANALYSIS OF CNF-SATISFIABILITY PROBLEM

## E.1  SATISFIABLE SITUATION

For example, similar to Eq. 3, let $\mathcal{C}_{\text{f}}$ : $\text{output}_{\text{f}} = A$ AND $B$ and $\mathcal{C}_{\text{r}}$ : $\text{output}_{\text{r}} = A$ OR $B$. Then,

$$\begin{aligned}
\Phi = &(\neg A \vee \neg B \vee \text{output}_{\text{f}}) \wedge (A \vee \neg\text{output}_{\text{f}}) \wedge (B \vee \neg\text{output}_{\text{f}}) \wedge (A \vee B \vee \neg\text{output}_{\text{r}}) \wedge \\
&(\neg A \vee \text{output}_{\text{r}}) \wedge (\neg B \vee \text{output}_{\text{r}}) \wedge (\neg\text{output}_{\text{f}}) \wedge (\text{output}_{\text{r}})
\end{aligned} \tag{11}$$

$\Phi$ being satisfiable would require the value assignment for $[A, B, \text{output}_{\text{f}}, \text{output}_{\text{r}}]$ to be either $[0, 1, 0, 1]$ or $[1, 0, 0, 1]$. Both of these outcomes ensure that $\mathcal{C}_{\text{f}}$ is corrupted while $\mathcal{C}_{\text{r}}$ is preserved. In this case, one of $A$ and $B$ must be changed while the other is preserved. For instance, if $A = 1$ and $B = 0$, preserving $A$ has no impact on violating $\mathcal{C}_{\text{f}}$ or maintaining $\mathcal{C}_{\text{r}}$. Therefore, $A$ is a retain node, and correspondingly, $B$ is a forget node. Conversely, if $A = 0$ and $B = 1$, then $A$ becomes the forget node and $B$ the retain node. This is intuitive: in $\mathcal{C}_{\text{f}}$, $A$ and $B$ are related by $AND$ logic, so neither $A$ nor $B$ exclusively or independently contains the information to be forgotten. Similarly, in $\mathcal{C}_{\text{r}}$, $A$ and $B$ are related by $OR$ logic, which means that both $A$ and $B$ individually have a significant influence on $\text{output}_{\text{r}}$.

## E.2  UNSATISFIABLE SITUATION

For example, let $\mathcal{C}_{\text{f}}$ : $\text{output}_{\text{f}} = A$ OR $B$ and $\mathcal{C}_{\text{r}}$ : $\text{output}_{\text{r}} = B$ AND $C$. Then,

$$\begin{aligned}
\Phi = &(A \vee B \vee \neg\text{output}_{\text{f}}) \wedge (\neg A \vee \text{output}_{\text{f}}) \wedge (\neg B \vee \text{output}_{\text{f}}) \wedge (\neg B \vee \neg C \vee \text{output}_{\text{r}}) \wedge \\
&(B \vee \neg\text{output}_{\text{r}}) \wedge (C \vee \neg\text{output}_{\text{r}}) \wedge (\neg\text{output}_{\text{f}}) \wedge (\text{output}_{\text{r}})
\end{aligned} \tag{12}$$

$\Phi$ being satisfiable would require the $\text{output}_{\text{f}} = 0$, and thus A$= 0$, B$= 0$. However, $\Phi$ would also require the $\text{output}_{\text{r}} = 1$, and thus B$= 1$, C$= 1$. In this instance, $A$ must be 0, which categorizes it

as a forget node. This is because the state of $A$ alone is sufficient to determine the outcome of $\mathcal{C}_f$, meaning $A$ exclusively and independently contains the information to be forgotten. Analogously, $C$ must be 1, identifying it as a retain node, as it is independent of $\mathcal{C}_f$ and thus requires no modification. $B$, however, is a conflict node: to remove $\mathcal{C}_f$, it must be changed (value=0), yet to preserve $\mathcal{C}_r$, it must be maintained ( value=1).

# F  DETAILS ABOUT EXPERIMENT SETUPS

## F.1  MODEL CONFIGURATIONS

For the WMDP task, we select the original Zephyr-7B-beta as the pretrained model. For the PKU-SafeRLHF task, we selected LLaMA2-7B as the foundational model for our study. All experiments were conducted on 16 NVIDIA RTX A100 GPUs. Each experiment takes approximately 5 minutes per 100 steps. We adopt the same rejection-based answers designed by Jia et al. (2024).

## F.2  UNLEARNING CONFIGURATIONS

All fine-tuning are conducted over 6 epochs, with 1 epoch for forget nodes and the rest 5 epochs for conflict nodes. The learning rate is grid-searched at $1 \times 10^{-5}$ for each task and datasets. The parameter $\lambda$ is set to 1 for each method across all tasks, and we adopted AdamW (Loshchilov & Hutter, 2017) as the optimizer.

## F.3  BASELINES

**DEPN** (Wu et al., 2023) (Detect and Edit Privacy Neurons) is a framework designed to safeguard against privacy leakage in pretrained language models by localizing and editing specific neurons. The method's core localization component is a novel privacy neuron detector that uses a gradient-based attribution technique. This detector computes a privacy attribution score for each neuron to quantify its contribution to the model's leakage of private information. This is achieved by calculating the cumulative gradient of the output probability with respect to the neuron's activation value, as the activation is gradually changed from zero to its original value.

**WAGLE** (Jia et al., 2024) (Weight Attribution-guided LLM Unlearning Framework) is a framework that pinpoints the most influential weights for unlearning through a strategic weight attribution method. The method frames the weight attribution problem as a bi-level optimization (BLO) problem, which allows it to balance unlearning efficacy with utility preservation. The core of the localization process is the derivation of a closed-form attribution score for each weight, calculated using the implicit gradient from the BLO problem. This score's value is determined by combining the gradients from both the forget loss and the retain loss.

**PCGU** (Yu et al., 2023) (Partitioned Contrastive Gradient Unlearning) is a gray-box method for unlearning social biases by localizing the specific weights responsible for encoding them. The method's localization strategy is based on comparing gradients from "contrastive sentence pairs," which are sentences that are minimally different in a specific domain, such as gender. PCGU first partitions the model's parameter set into discrete weight vectors or blocks. It then computes the gradients for each sentence in a pair with respect to these weight blocks. By measuring the cosine similarity between the gradients of the two sentences, it identifies the weight blocks that are most relevant to the targeted bias (i.e., those with the lowest cosine similarity between their gradients).

**MEMIT** (Patil et al., 2023) ((Mass-Editing Memory in a Transformer)) addresses the deletion of factual information by causal tracing, a denoising-based intervention method. This approach relies on the assumption that knowledge is stored in specific, localized components of the network, and can be identified via causal mediation.

**GA** (Yao et al., 2024) (Gradient Ascent) encourages the response of the LLM post-unlearning to deviate from its original response within the training set.

**NPO** (Zhang et al., 2024) (Negative Preference Optimization) specifies the forget loss as the loss of direct preference optimization by treating the forgotten data exclusively as negative examples. The

NPO loss outperforms the GA loss due to its improved stability, avoiding catastrophic collapse in forgetting and utility preservation during optimization.

**PO** (Maini et al., 2024) (Preference Optimization) is also inspired by DPO but introduces targeted unlearning responses such as 'I don't know' or responses stripped of sensitive information, treating these exclusively as positive examples for preference alignment.

### F.4 RETAIN SET

We select a series of specific tasks as retain set: Winogrande, SST-2, RTE, Bool, Induction, IOI, Gender Bias, Docstring, Great Than, SA, arithmetic, Reverse. We show the examples of each task in the Table 4.

Table 4: An overview of the datasets of specific tasks.

| Task | Example | Label |
|---|---|---|
| Winograde | John moved the couch from the garage to the backyard to create space. The _ is small. | garage |
| SST-2 | hide new secretions from the parental units | negative |
| RTE | No Weapons of Mass Destruction Found in Iraq Yet.
Weapons of Mass Destruction Found in Iraq. | not entailment |
| Bool | (True AND True) OR False | True |
| Induction | Vernon Dursley and Petunia Durs | ley |
| IOI | When John and Mary went to the store, Mary gave a bottle of milk to | John |
| Gender Bias | So Evan is a really great friend, isn't | he |
| Docstring | def f(self, files, obj, state, size, shape, option):
:param state: performance analysis
:param size: pattern design
:param | shape |
| Great Than | The war lasted from 1517 to 15 | 18 |
| SA | Many girls insulted | themselves |
| arithmetic | 12 plus 18 equals | 30 |
| Reverse | [0, 3, 2, 1] | [1, 2, 3, 0] |

## G RESULTS ON DETAILED GENERAL UTILITY

In this section, we report the specific accuracy for all non-target tasks, which can be found in Table 5.

Table 5: Performance on specific non-target tasks. We report the accuracy metric with WMDP Cyber as forget set.

| retain set | Method | mmlu | $arc_{challenge}$ | $arc_{easy}$ | boolq | hellawasg | openbookqa | piqa | rte | $truthqa_{gen}$ | $truthqa_{mc1}$ | $truthqa_{mc2}$ | winogrande | average |
|---|---|---|---|---|---|---|---|---|---|---|---|---|---|---|
| Winogrande | GA | 0.2655 | 0.2144 | 0.3214 | 0.5417 | 0.2549 | 0.202 | 0.6041 | 0.4571 | 0.0635 | 0.2216 | 0.4371 | 0.4019 | 0.3322 |
| | NPO | 0.2647 | 0.2194 | 0.3317 | 0.5549 | 0.2517 | 0.202 | 0.6044 | 0.4419 | 0.0571 | 0.2549 | 0.3517 | 0.4219 | 0.3296 |
| | PO | 0.2846 | 0.2519 | 0.3946 | 0.5617 | 0.2207 | 0.2407 | 0.5817 | 0.5841 | 0.0617 | 0.2519 | 0.4217 | 0.5517 | 0.3688 |
| | MEMIT | 0.2846 | 0.2573 | 0.3907 | 0.5519 | 0.2517 | 0.2419 | 0.5719 | 0.6671 | 0.0938 | 0.2594 | 0.4417 | 0.6074 | 0.3847 |
| | PCGU | 0.2847 | 0.2674 | 0.3847 | 0.5671 | 0.2694 | 0.2574 | 0.5571 | 0.6574 | 0.0473 | 0.3097 | 0.4571 | 0.533 | 0.3827 |
| | DEPN | 0.2501 | 0.2857 | 0.4184 | 0.6217 | 0.401 | 0.226 | 0.6893 | 0.6282 | 0.3293 | 0.2289 | 0.4616 | 0.6283 | 0.4314 |
| | WAGLE | 0.3337 | 0.3336 | 0.5543 | 0.8116 | 0.3868 | 0.18 | 0.6757 | 0.7292 | 0.033 | 0.2387 | 0.4563 | 0.5722 | 0.4421 |
| | CLUE | 0.4104 | 0.3387 | 0.5762 | 0.8367 | 0.3785 | 0.202 | 0.6474 | 0.6859 | 0.0612 | 0.2583 | 0.4926 | 0.6101 | 0.4582 |
| SST-2 | GA | 0.2144 | 0.2549 | 0.4127 | 0.3147 | 0.2571 | 0.1256 | 0.4679 | 0.5217 | 0.0214 | 0.2264 | 0.4172 | 0.5174 | 0.3129 |
| | NPO | 0.2347 | 0.2617 | 0.4318 | 0.3604 | 0.3618 | 0.1304 | 0.4729 | 0.5329 | 0.0317 | 0.2517 | 0.4237 | 0.5367 | 0.3361 |
| | PO | 0.2509 | 0.2847 | 0.4137 | 0.4097 | 0.3849 | 0.1517 | 0.4593 | 0.5873 | 0.0437 | 0.2849 | 0.4307 | 0.5376 | 0.3528 |
| | MEMIT | 0.2617 | 0.2849 | 0.4219 | 0.4137 | 0.2849 | 0.2576 | 0.4581 | 0.5837 | 0.0674 | 0.2947 | 0.4137 | 0.5643 | 0.3591 |
| | PCGU | 0.2517 | 0.2719 | 0.4739 | 0.4016 | 0.2674 | 0.2519 | 0.4673 | 0.5917 | 0.0419 | 0.2873 | 0.4473 | 0.5879 | 0.3622 |
| | DEPN | 0.3047 | 0.2617 | 0.5493 | 0.3677 | 0.4019 | 0.22 | 0.6579 | 0.5017 | 0.0463 | 0.2017 | 0.4219 | 0.6257 | 0.3796 |
| | WAGLE | 0.3114 | 0.2807 | 0.4491 | 0.3899 | 0.3931 | 0.22 | 0.6464 | 0.5207 | | 0.2497 | 0.4664 | 0.6582 | 0.3844 |
| | CLUE | 0.2387 | 0.2901 | 0.436 | 0.3914 | 0.4192 | 0.22 | 0.642 | 0.5451 | 0.0465 | 0.2668 | 0.4985 | 0.6622 | 0.3880 |
| RTE | GA | 0.2017 | 0.2347 | 0.4419 | 0.6057 | 0.3617 | 0.1208 | 0.5319 | 0.4739 | 0.207 | 0.2057 | 0.4317 | 0.5976 | 0.3677 |
| | NPO | 0.2067 | 0.2217 | 0.4319 | 0.2977 | 0.3517 | 0.1549 | 0.5037 | 0.4497 | 0.202 | 0.1849 | 0.4395 | 0.63517 | 0.3394 |
| | PO | 0.2149 | 0.2549 | 0.4367 | 0.6207 | 0.3491 | 0.1422 | 0.5537 | 0.5037 | 0.202 | 0.2067 | 0.4019 | 0.6082 | 0.3745 |
| | MEMIT | 0.2166 | 0.2537 | 0.4691 | 0.6255 | 0.3847 | 0.1437 | 0.5594 | 0.5017 | 0.217 | 0.2257 | 0.4137 | 0.6107 | 0.3864 |
| | PCGU | 0.2147 | 0.2549 | 0.4317 | 0.6217 | 0.3429 | 0.147 | 0.5517 | 0.5037 | 0.2094 | 0.2037 | 0.4067 | 0.6071 | 0.3755 |
| | DEPN | 0.2547 | 0.302 | 0.4701 | 0.6584 | 0.375 | 0.184 | 0.6643 | 0.5487 | 0.2367 | 0.2264 | 0.4482 | 0.633 | 0.4168 |
| | WAGLE | 0.3784 | 0.2834 | 0.4377 | 0.5957 | 0.3365 | 0.196 | 0.6268 | 0.7329 | 0.0257 | 0.2558 | 0.4963 | 0.6006 | 0.4138 |
| | CLUE | 0.4074 | 0.2722 | 0.607 | 0.5667 | 0.4339 | 0.18 | 0.6115 | 0.7354 | 0.0808 | 0.2497 | 0.4922 | 0.602 | 0.4366 |
| Bool | GA | 0.2176 | 0.2517 | 0.3744 | 0.2849 | 0.3479 | 0.1437 | 0.4873 | 0.4319 | 0.207 | 0.1673 | 0.4037 | 0.6037 | 0.3257 |
| | NPO | 0.2943 | 0.2674 | 0.4237 | 0.3619 | 0.3729 | 0.2046 | 0.6273 | 0.5037 | 0.0237 | 0.2219 | 0.4439 | 0.6319 | 0.3655 |
| | PO | 0.3114 | 0.2849 | 0.4437 | 0.3958 | 0.3946 | 0.2344 | 0.6491 | 0.5267 | 0.0255 | 0.2438 | 0.4691 | 0.6533 | 0.3841 |
| | MEMIT | 0.2855 | 0.2515 | 0.4973 | 0.7067 | 0.2943 | 0.127 | 0.5937 | 0.5217 | 0.1247 | | 0.359 | 0.5438 | 0.3867 |
| | PCGU | 0.3057 | 0.2294 | 0.4538 | 0.6471 | 0.2594 | 0.106 | 0.5937 | 0.5938 | 0.2811 | 0.1673 | 0.3894 | 0.4936 | 0.3769 |
| | DEPN | 0.2935 | 0.2527 | 0.6199 | 0.5973 | 0.2867 | 0.196 | 0.4937 | 0.6273 | 0.418 | 0.1579 | 0.3657 | 0.4533 | 0.3964 |
| | WAGLE | 0.2691 | 0.2517 | 0.3784 | 0.7055 | 0.3259 | 0.19 | 0.6007 | 0.5776 | 0.033 | 0.2509 | 0.4837 | 0.5833 | 0.3875 |
| | CLUE | 0.3706 | 0.3157 | 0.4949 | 0.7324 | 0.3737 | 0.206 | 0.6627 | 0.5848 | 0.0747 | 0.2521 | 0.4893 | 0.6551 | 0.4343 |

## H  RESULTS ON MIA AND ROUGE-L

As introduced by Jia et al. (2024), membership inference attack (MIA) is evaluated by the area under the ROC curve using Min-20% Prob to detect if the provided text belongs to the training or testing set. We apply MIA to the forget set; thus, a higher MIA score indicates a higher confidence in predicting that the forget data point does not belong to the training set. Moreover, Rouge-L recall is also measured over the forget set. A lower value corresponds to better unlearning. The metric 1-Rouge-L is also used for ease of performance averaging. We show the results of MIA and Rouge-L in Table 6.

Table 6: Performance overview of LLM unlearning with 1-accuracy, MIA, Rouge-L as metrics for forget efficacy.

| Method | Unlearned Parameter | Winogrande | | | SST-2 | | | RTE | | | Bool | | |
|---|---|---|---|---|---|---|---|---|---|---|---|---|---|
| | | 1-accuracy↑ | MIA↑ | Rouge-L↑ | 1-accuracy↑ | MIA↑ | Rouge-L↑ | 1-accuracy↑ | MIA↑ | Rouge-L↑ | 1-accuracy↑ | MIA↑ | Rouge-L↑ |
| **WMDP Cyber** | | | | | | | | | | | | | |
| Origin | - | 0.4454 | 0.4238 | 0.0159 | 0.4454 | 0.4394 | 0.0159 | 0.4454 | 0.4163 | 0.0159 | 0.4454 | 0.4361 | 0.0159 |
| GA | 100% | 0.6583 | 0.9517 | 0.3957 | 0.6651 | 0.9428 | 0.3849 | 0.6477 | 0.9257 | 0.3829 | 0.6527 | **0.9637** | 0.3915 |
| NPO | 100% | 0.6639 | **0.9647** | 0.3296 | 0.6542 | **0.9556** | 0.3511 | 0.6976 | **0.9645** | 0.3519 | 0.6548 | 0.9428 | 0.3156 |
| PO | 100% | 0.6851 | 0.6357 | 0.3519 | 0.6729 | 0.6724 | 0.3691 | 0.6851 | 0.5821 | 0.3636 | 0.6826 | 0.6411 | 0.3894 |
| MEMIT | 76.27% | 0.6691 | 0.6259 | 0.4029 | 0.6738 | 0.6619 | 0.3664 | 0.6759 | 0.6237 | 0.3594 | 0.6908 | 0.6258 | 0.3674 |
| PCGU | 86.77% | 0.6724 | 0.6871 | 0.4336 | 0.6721 | 0.6849 | 0.3294 | 0.6691 | 0.6482 | 0.3667 | 0.6955 | 0.6364 | 0.3845 |
| DEPN | 78.82% | 0.6955 | 0.6644 | 0.4418 | 0.7025 | 0.6237 | 0.3558 | 0.7129 | 0.6553 | 0.3127 | 0.7156 | 0.6318 | 0.3946 |
| WAGLE | 90.01% | **0.7021** | 0.6821 | 0.4309 | 0.7081 | 0.6884 | 0.3619 | 0.6851 | 0.6418 | 0.3618 | 0.7217 | 0.6138 | 0.3746 |
| CLUE | **58.16%** | 0.6975 | 0.7926 | **0.4692** | **0.7333** | 0.7713 | **0.4671** | **0.7445** | 0.7827 | **0.4967** | 0.7242 | 0.7734 | **0.4108** |
| **WMDP Bio** | | | | | | | | | | | | | |
| Origin | - | 0.3551 | 0.4109 | 0.0122 | 0.3551 | 0.4219 | 0.0122 | 0.3551 | 0.4057 | 0.0122 | 0.3551 | 0.4577 | 0.0122 |
| GA | 100% | 0.5647 | **0.9662** | 0.3755 | 0.6751 | 0.9554 | 0.2741 | 0.5649 | 0.9234 | 0.3685 | 0.5683 | 0.9384 | 0.3815 |
| NPO | 100% | 0.5718 | 0.9517 | 0.3215 | 0.6719 | **0.9618** | 0.3138 | 0.5741 | **0.9543** | 0.3348 | 0.5722 | **0.9613** | 0.2348 |
| PO | 100% | 0.6059 | 0.6349 | 0.3851 | 0.6853 | 0.6138 | 0.2348 | **0.5892** | 0.6518 | 0.4318 | 0.5856 | 0.6138 | 0.4128 |
| MEMIT | 74.29% | 0.5919 | 0.5647 | 0.3189 | 0.6955 | 0.5196 | 0.4318 | 0.5477 | 0.6618 | 0.3841 | 0.6051 | 0.6138 | 0.3188 |
| PCGU | 85.12% | 0.6011 | 0.6219 | 0.4318 | 0.6849 | 0.6138 | 0.313 | 0.5391 | 0.5384 | 0.3186 | 0.5967 | 0.3561 | 0.3388 |
| DEPN | 77.24% | 0.6053 | 0.6358 | 0.3189 | 0.7018 | 0.6038 | 0.4318 | 0.5463 | 0.6138 | 0.3181 | 0.5993 | 0.5313 | 0.4885 |
| WAGLE | 90.02% | 0.5997 | 0.6617 | 0.4189 | 0.6984 | 0.5831 | 0.4831 | 0.5491 | 0.6913 | 0.3384 | 0.6009 | 0.6318 | 0.4528 |
| CLUE | **56.19%** | 0.6174 | 0.6922 | **0.4851** | 0.7136 | 0.7216 | **0.6599** | 0.5869 | 0.6955 | **0.5219** | 0.6123 | 0.6419 | **0.6335** |
| **PKU-SafeRLHF** | | | | | | | | | | | | | |
| origin | - | 0.2941 | 0.4219 | 0.0094 | 0.2941 | 0.4655 | 0.0094 | 0.2941 | 0.4319 | 0.0094 | 0.2941 | 0.4408 | 0.0094 |
| GA | 100% | 0.6154 | 0.9217 | 0.3574 | 0.6055 | **0.9517** | 0.3519 | 0.6259 | **0.9492** | 0.3622 | 0.6019 | 0.9247 | 0.3661 |
| NPO | 100% | 0.6055 | **0.9315** | 0.3691 | 0.6129 | 0.9427 | 0.3367 | 0.6144 | 0.9255 | 0.3359 | 0.6168 | **0.9366** | 0.2943 |
| PO | 100% | 0.6259 | 0.6319 | 0.4296 | 0.6235 | 0.6217 | 0.4219 | 0.6238 | 0.6731 | 0.4127 | 0.6255 | 0.8412 | 0.5137 |
| MEMIT | 77.62% | 0.6459 | 0.5839 | 0.4935 | 0.6491 | 0.5394 | 0.4339 | 0.6255 | 0.6329 | 0.3629 | 0.6471 | 0.6719 | 0.3233 |
| PCGU | 86.29% | 0.6394 | 0.5638 | 0.4163 | 0.6336 | 0.6173 | 0.3659 | 0.6399 | 0.5843 | 0.3816 | 0.6392 | 0.6652 | 0.4062 |
| DEPN | 74.36% | 0.6617 | 0.6173 | 0.3195 | 0.6573 | 0.6628 | 0.3816 | 0.6347 | 0.6319 | 0.4457 | 0.6429 | 0.5937 | 0.4138 |
| WAGLE | 90.01% | 0.6559 | 0.6284 | 0.3326 | 0.6637 | 0.5973 | 0.4219 | 0.6417 | 0.6642 | 0.4369 | 0.6357 | 0.6173 | 0.3907 |
| CLUE | **54.88%** | **0.7249** | 0.8411 | **0.6305** | **0.6813** | 0.7359 | **0.6262** | **0.6561** | 0.6904 | **0.5391** | **0.6594** | 0.7216 | **0.6617** |

## I  SCALABILITY AND ROBUSTNESS OF CLUE AT VARIOUS FORGET RATIOS

In this section, we investigate the performance of CLUE and other localization methods under varying forget ratios. Specifically, we define the **forget ratio** as the ratio of modified parameters and examine the performance of our localization methods in terms of forget efficacy, retain utility, and general utility for forget ratios of [0.1, 0.2, 0.5, 0.8, 0.9]. For our experiments, the retain set used is SST-2.

Figure 7 shows the performance on WMDP Cyber, WMDP Bio, and PKU-safeRLHF. It is evident that as the forget ratio increases, the unlearning efficacy improves; however, the utility trend is more volatile and non-monotonic. Nevertheless, CLUE consistently outperforms other localization methods. This demonstrates that our identification of both forget nodes and conflict nodes is beneficial for the unlearning task across all tested forget parameter ratios.

## J  DISTRIBUTION OF DIFFERENT NODES

In this section, we investigate the proportion of different parameter types within forget nodes and conflict nodes. For a comparative analysis, we also include the parameter distributions from MEMIT and WAGLE. MEMIT uses a method similar to circuit discovery to identify important nodes but relies exclusively on a denoising-based intervention. As has been confirmed by existing work (Chen et al., 2025), such a circuit is incomplete, lacking the necessary logical reasoning nodes. WAGLE, on the other hand, employs a gradient-based weight attribution method, which often makes it difficult to discover equivalent paths within OR gates.

Figure 8 illustrates the node distributions for four retain sets, with the Zephyr-7B-beta model and the WMDP Cyber dataset serving as the forget set. MLPs constitute the largest proportion, as

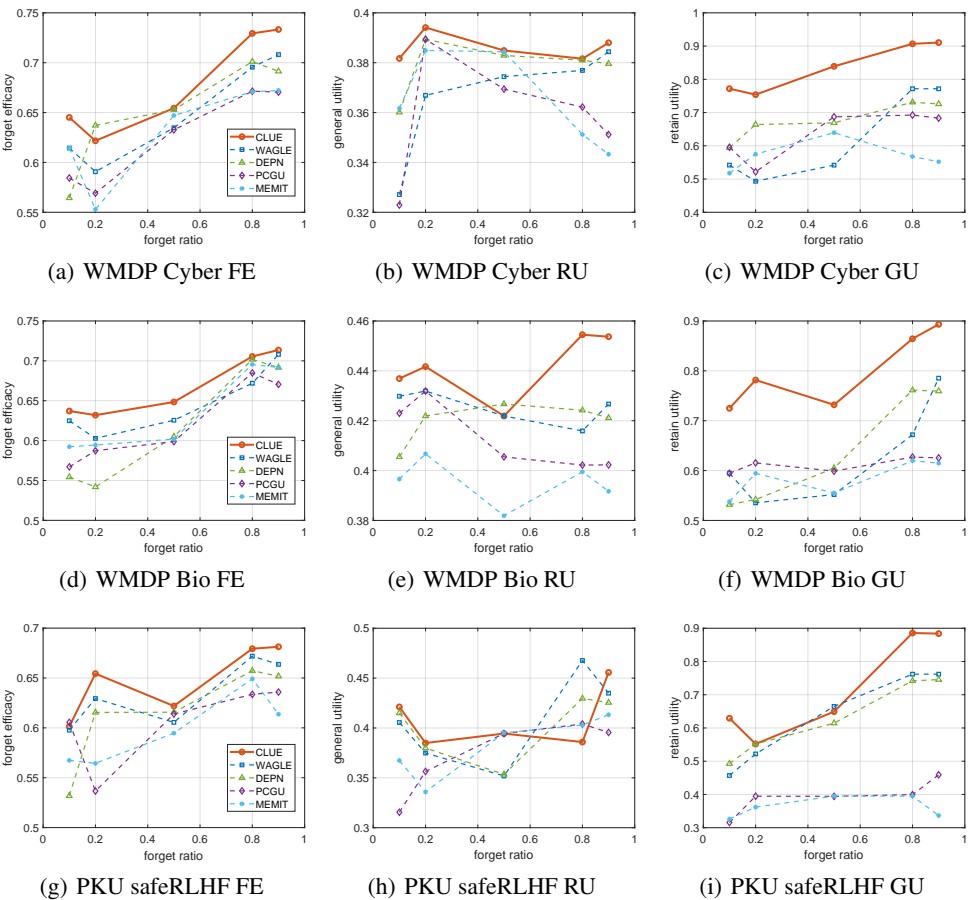

Figure 7: Preference of LLM unlearning at various forget ratios, FE means forget efficacy, RU represents the retain utility, GU represents general utility.

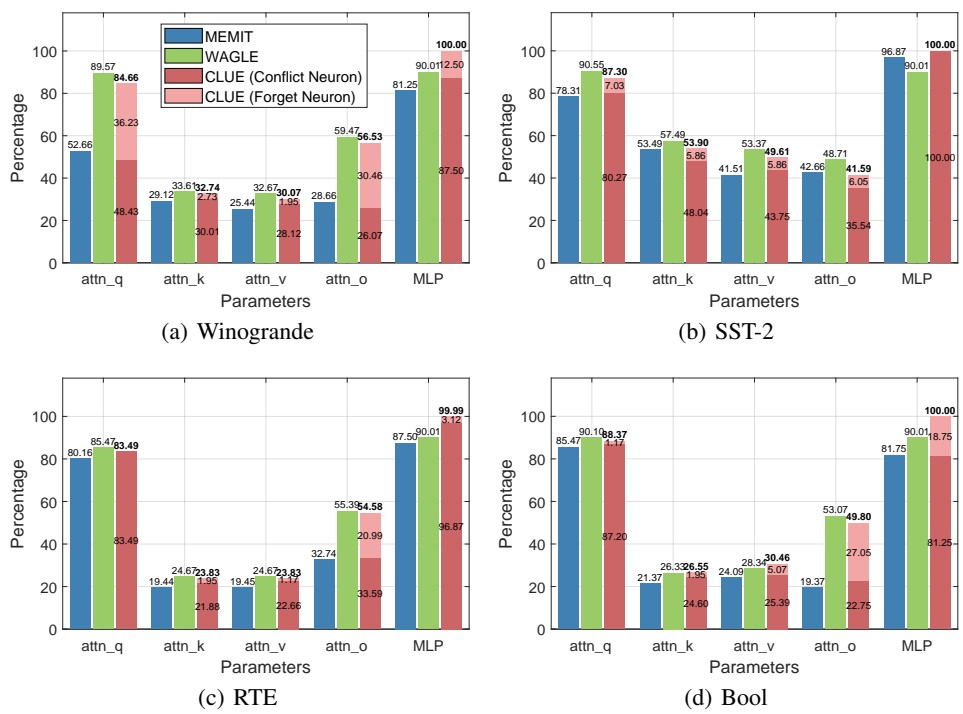

Figure 8: Nodes distribution of Zephyr-7B-beta model in WMDP Cyber.

they are generally considered the most information-rich memory units. An interesting pattern also emerges: outside of the MLPs, the number of nodes identified by MEMIT is similar to the number of conflict nodes found by our method (CLUE). In contrast, the number of nodes found by WAGLE is comparable to the sum of both forget nodes and conflict nodes identified by our approach. This observation further confirms our viewpoint. Due to its lack of circuit completeness, MEMIT fails to discover forget nodes from a sufficient logical structure. Meanwhile, WAGLE, by not considering the influence of causal effects, cannot discover OR gates and thus includes all common nodes.

## K  THE BENCHMARK OF COMPUTATION COST

To evaluate the time and computational resource expenditure of our method, particularly in comparison to existing approaches, we conducted assessments on the WMDP Cyber and WMDP Bio datasets using the Zephyr-7B-beta model. The experimental environment was equipped with 16 NVIDIA RTX A100 GPUs. For comparison with existing methods, we selected GA as a representative fine-tuning method and WAGLE as a representative localization method. The Table 7 and  8 shows the time required for localization and fine-tuning for these methods. It demonstrates that our method does not introduce significant time overhead. However, when combined with the performance in forget efficacy and retain utility presented in Table 1, our method exhibits the highest cost-effectiveness. Specifically, we evaluated the computational time of our approach on two circuit discovery methods: Edge-Pruning, which fits the circuit by adding a differentiable mask, and EAP, which directly computes the circuit using a first-order Fourier transform. Furthermore, the time required for SAT solving was consistently under one minute, largely attributable to a well-defined initialization process. Consequently, the overall computational overhead of our entire pipeline is comparable to that of existing approaches. In particular, when EAP is employed as the circuit discovery method, the runtime is significantly lower than the average of current methods.

Table 7: Computation Cost in WMDP Cyber.

| Method | Localization | | | Fine-tuning | All |
|---|---|---|---|---|---|
| | Circuit discovery | SAT Solving | All | | |
| GA | - | - | - | 4.8h | 4.8h |
| WAGLE | - | - | 0.5h | 3.2h | 3.7h |
| Ours (Edge-Pruning) | 2.1h | 0.01h | 2.11h | 3.1h | 5.21h |
| Ours (EAP) | 0.1h | 0.01h | 0.11h | 3.1h | 3.21h |

Table 8: Computation Cost in WMDP Bio.

| Method | Localization | | | Fine-tuning | All |
|---|---|---|---|---|---|
| | Circuit discovery | SAT Solving | All | | |
| GA | - | - | - | 3.5h | 3.5h |
| WAGLE | - | - | 0.4h | 2.9h | 3.3h |
| Ours (Edge-Pruning) | 1.7h | 0.01h | 1.71h | 2.9h | 3.61h |
| Ours (EAP) | 0.1h | 0.01h | 0.11h | 2.9h | 3.01h |

