# OpenReview forum: "CLUE: Conflict-guided Localization for LLM Unlearning Framework"
_ICLR.cc/2026/Conference — ICLR 2026 Poster_

### Official Review · Reviewer_J9hV · 2025-10-26

**Soundness:** 3
**Presentation:** 3
**Contribution:** 3
**Rating:** 6
**Confidence:** 3

**Summary:**

The paper introduces CLUE, a new framework designed to improve the precision and effectiveness of large language model (LLM) unlearning. Existing localization-based unlearning methods identify important neurons for modification but fail to distinguish between neurons responsible for forgetting and those critical for retention. This entanglement often leads to over-forgetting or incomplete erasure of target information. To solve this, CLUE leverages circuit discovery, a mechanistic interpretability approach, to map neuron interactions as circuits. It then converts these into CNF and uses Boolean satisfiability solving to disentangle neurons into three categories: Forget neurons, Retain neurons, and Conflict neurons.

**Strengths:**

- The application of mechanistic interpretability and circuit discovery to LLM unlearning is novel and conceptually well-motivated, bridging interpretability with model editing.
- The explicit separation into forget, retain, and conflict neurons allows for more precise control and understanding of unlearning behavior.
- Experiments across multiple benchmarks show consistent improvements.
- The framework enhances interpretability by revealing how specific neurons and circuits contribute to different functions, aligning with future modular LLM design goals.
- The paper clearly articulates the problem of neuron entanglement and justifies the need for a more granular approach like CLUE.

**Weaknesses:**

- Although the section 5.1 and 5.2 documented well on benchmarks that the study is evaluted on plus the details on how the evaluation is conducted, I, somehow, could not find any descriptions on what model is evaluated. Is it the same model across all the benchmark evaluations? What is the family and the size of the model being evaluated in the study?
- As the curse of circuit discovery, the framework would possibly also have scalability problem, such as being computationally expensive to find a meaningful ciruit in larger models. For meaningful circuit, I mean a circuit that is sparse enough comparing to the original computational graph.
- The framework utilizes the edge pruning technique. In edge pruning, it returns a series of ciruits as the sparisity going down, while the metrics of the faithfulness would also vary. I wonder what are the specific critierion that was used to pick such circuit from the edging pruning outcomes. And how would you justify picking one of the circuits over the others ( maybe in terms of exact match, KL divergence, or sparisity)?
- The framework appears to make an implicit assumption that the contribution of each neuron identified within a circuit is independent of the specific path or connectivity structure it participates in. For instance, consider a sequence of neurons
𝐴1→𝐴2→𝐴3 across successive layers. CLUE might categorize 𝐴1, 𝐴2, and 𝐴3 into different neuron sets (e.g., forget, retain, or conflict) based on their individual CNF assignments. However, in reality, the functional contribution of each neuron is context-dependent on the path it forms with others. In contrast, circuit-level methods such as edge pruning often treat  A1→A2 →A3 as an integrated substructure, preserving or removing the path as a whole. This suggests that CLUE’s neuron-wise localization might overlook interdependent effects among neurons within the same circuit, potentially leading to suboptimal or fragmented unlearning.

**Questions:**

Please refer to the weakness section.

---

> ### Author Response · Authors · 2025-11-14
> **Response**
>
> We appreciate your positive feedback on our paper and thank you for your meticulous and profound questions. We are happy to provide our responses to each of them below.
>
> ### Q1: Although the section 5.1 and 5.2 documented well on benchmarks that the study is evaluted on plus the details on how the evaluation is conducted, I, somehow, could not find any descriptions on what model is evaluated. Is it the same model across all the benchmark evaluations? What is the family and the size of the model being evaluated in the study?
>
> We appreciate your questions. The models we employed are the Zephyr-7B-beta model for WMDP Cyber and WMDP Bio, and LLaMA2-7B for PKU-SafeRLHF. We have specified this in **Section 5.1, lines 291-293**. Furthermore, these models are the original recommended models for their respective forget datasets.
>
> ### Q2: As the curse of circuit discovery, the framework would possibly also have scalability problem, such as being computationally expensive to find a meaningful ciruit in larger models. For meaningful circuit, I mean a circuit that is sparse enough comparing to the original computational graph.
> We appreciate your comment and must indeed acknowledge the scalability problem of circuit discovery, although some current research [1][2] is already exploring faster algorithms for larger-scale circuit discovery, and the logical circuit framework we employ is perfectly adaptable to these new algorithms. We will incorporate a discussion of this issue into the **limitation section** of revised version.
>
> ### Q3: The framework utilizes the edge pruning technique. In edge pruning, it returns a series of ciruits as the sparisity going down, while the metrics of the faithfulness would also vary. I wonder what are the specific critierion that was used to pick such circuit from the edging pruning outcomes. And how would you justify picking one of the circuits over the others ( maybe in terms of exact match, KL divergence, or sparisity)?
> We appreciate your questions. In **Section 6.3**, we reported the relationship between circuit sparsity, faithfulness, and unlearning performance. Regarding the optimal circuit selection, the results in Figure 5 empirically demonstrate that when the sparsity reaches **0.7**, the unlearning performance generally achieves the optimal range. Therefore, we selected circuits with a sparsity of 0.7 by default for all other experiments.
>
> ### Q4: The framework appears to make an implicit assumption that the contribution of each neuron identified within a circuit is independent of the specific path or connectivity structure it participates in. For instance, consider a sequence of neurons 𝐴1→𝐴2→𝐴3 across successive layers. CLUE might categorize 𝐴1, 𝐴2, and 𝐴3 into different neuron sets (e.g., forget, retain, or conflict) based on their individual CNF assignments. However, in reality, the functional contribution of each neuron is context-dependent on the path it forms with others. In contrast, circuit-level methods such as edge pruning often treat A1→A2 →A3 as an integrated substructure, preserving or removing the path as a whole. This suggests that CLUE’s neuron-wise localization might overlook interdependent effects among neurons within the same circuit, potentially leading to suboptimal or fragmented unlearning.
>
> Thank you for your question. In fact, you do not need to be concerned about such an occurrence. Using the example you mentioned, $A_1 \rightarrow A_2 \rightarrow A_3$, let us assume all three belong to the retain circuit. If we were to classify $A_1$ as a forget neuron, then there must exist an $A_1'$ that forms an OR gate with $A_1$. That is, a structure $A_1 \rightarrow A_2 \leftarrow A_1'$ exists, where $A_2 = A_1 \text{ OR } A_1'$.
>
> Furthermore, even if $A_1$ is classified as a conflict neuron, the dependency relationships from its parent nodes will still be transferred. (There is a **potential lemma** here: when a conflict appears in a child node, a conflict must also exist among its parent nodes.) Therefore, a chain relationship will invariably persist, connecting these neurons.
>
> In **summary**, you need not worry that the SAT solving of the circuit will completely disrupt path dependencies. In other words, the SAT solving process only separates those paths that can be separated (such as redundant or alternative paths within an OR gate), while the necessary and irreplaceable paths are still fully classified and preserved.
>
>
> reference:
>
> [1]. *Does circuit analysis interpretability scale? evidence from multiple choice capabilities in chinchilla, arxiv 2023*
>
> [2]. *Attribution Patching Outperforms Automated Circuit Discovery, Neurips workshop 2023*

---

> ### Comment · Reviewer_J9hV · 2025-11-14
>
> Thank you for the responses! My concerns are mostly addressed, and will maintain my score.

---

### Official Review · Reviewer_tdd5 · 2025-10-26

**Soundness:** 3
**Presentation:** 3
**Contribution:** 3
**Rating:** 6
**Confidence:** 5

**Summary:**

This paper addresses the challenge of LLM unlearning removing undesirable knowledge while preserving unrelated capabilities. The authors note that existing localization based methods, such as WAGLE and MEMIT, struggle to disentangle neurons responsible for forgetting from those involved in retaining information. To overcome this limitation, they propose CLUE (Conflict-Guided Localization for LLM Unlearning), a framework that integrates circuit discovery and logical reasoning. Specifically, CLUE first uses circuit discovery to extract distinct “forget” and “retain” subcircuits, then transforms these circuits into Conjunctive Normal Form (CNF) using the Tseitin transformation. The CNF satisfiability problem is subsequently solved to classify neurons into three categories: forget, retain, and conflict. Finally, the framework applies a two-stage fine tuning strategy first fine-tuning the forget neurons using a forget loss, followed by fine-tuning the conflict neurons with both forget and retain losses. Experimental results on benchmark datasets such as WMDP-Cyber, WMDP-Bio, and PKU-SafeRLHF, using Zephyr-7B and LLaMA2-7B models, demonstrate that CLUE achieves superior forgetting efficacy and retain utility while modifying significantly fewer parameters than previous methods like GA, NPO, PO, MEMIT, DEPN, and WAGLE.

**Strengths:**

Clear theoretical grounding and stepwise formulation (Eq. 1–6).
Comprehensive experiments across tasks and metrics (1-Acc, MIA, ROUGE-L).
Detailed ablations isolating the effect of forget/conflict masks and finetuning strategies.
Insightful analysis of sparsity vs. faithfulness (Fig. 4) and neuron-type shifts post unlearning (Table 3).
Public anonymized code and strong reproducibility statement.

**Weaknesses:**

Circuits are treated statically; dynamic updates during finetuning could better reflect neuron shifts.
Computational overhead of SAT solving and circuit extraction is not benchmarked.
Limited discussion on scaling to models > 7 B parameters.
Retain set selection sensitivity deserves deeper analysis (beyond Figure 3 trends).
Language polishing could improve readability.

**Questions:**

For multiple retain sets, could a multi-objective SAT formulation handle overlapping conflicts more gracefully?
Is there a failure case where CNF becomes unsatisfiable for large overlapping circuits, and how is it handled?

---

> ### Author Response · Authors · 2025-11-14
> **Response**
>
> Thank you for your positive feedback and insightful suggestions on our manuscript. We appreciate your professional comments and have addressed them one by one below:
>
> ### Q1: Circuits are treated statically; dynamic updates during finetuning could better reflect neuron shifts.
> Thank you for your suggestion. We are in complete agreement with your viewpoint. In fact, we have already emphasized this in the **limitations section**:
> > "The circuit is static and cannot dynamically reflect changes in key neurons during the fine-tuning process. Therefore, exploring the dynamics of circuits during parameter fine-tuning is a direction for future work. "
>
> Synchronously updating the circuit during fine-tuning would indeed be a superior process. However, the time and computational resource expenditure this would entail is unacceptable. Fortunately, we have demonstrated that significant superiority can be achieved even by localizing a static circuit. Consequently, we have designated the issue of dynamic circuits as a primary focus for future research.
>
> ### Q2: Computational overhead of SAT solving and circuit extraction is not benchmarked.
> We appreciate your comment. We have now added **Appendix J** to provide a brief comparison of the computational time of our method against that of other existing approaches. In brief, when employing the fastest circuit discovery algorithm, our method does not significantly increase the time expenditure or require additional computational resources.
>
> ### Q3: Limited discussion on scaling to models > 7 B parameters.
> We appreciate your comment and concede that this is indeed a limitation. However, it is worth noting that within the current unlearning research, models at the 7B scale have become the **predominant size** for evaluation. Therefore, while we acknowledge the lack of exploration on larger-scale models, we believe that the 7B size is sufficient for validating the unlearning process.
>
> ### Q4: Retain set selection sensitivity deserves deeper analysis (beyond Figure 3 trends).
> Thank you for your suggestion. The analysis of multiple retain sets, particularly those of different types, is indeed a direction worthy of exploration, and we have emphasized this point in the **limitations section**.
>
> However, current methods are constrained by the uniformity of gradient backpropagation, making it difficult to provide feedback for individual retain sets in a multi-retain-set scenario. Therefore, we are currently exploring direct parameter editing methods, similar to those currently prevalent in the field of knowledge editing[1]. This, however, is beyond the scope of the present paper.
>
> ### Q5: Language polishing could improve readability.
> We appreciate your suggestion. We have updated the definitions of certain terms and the explanations of key processes within the article. Specifically, you may refer to the revisions we have summarized in the **General Response**. We hope these modifications enhance the clarity and comprehensibility of our manuscript.
>
> ### Q6: For multiple retain sets, could a multi-objective SAT formulation handle overlapping conflicts more gracefully? Is there a failure case where CNF becomes unsatisfiable for large overlapping circuits, and how is it handled?
> We appreciate your questions. In scenarios involving multiple retain sets, we are able to significantly reduce unnecessary solving iterations through effective **initialization**. For instance, neurons within an AND gate in the retain circuit can be directly set to 1 during initialization, which greatly curtails the solving process required for conflict neurons.
>
> Furthermore, based on current experimental results, we have not yet encountered a solver collapse resulting from large overlapping circuits. As indicated by the overlap results presented in **Table 3**, it is **highly improbable** for the forget set and the retain set to generate a **large-scale overlap**. (By definition, if a substantial overlap were to occur, it would imply that the sets are difficult to distinguish, thus making it challenging to guarantee retain utility while simultaneously achieving forget efficacy.)
>
>
> reference:
>
> [1]. *AlphaEdit: Null-Space Constrained Knowledge Editing for Language Models, ICLR 2025*

---

### Official Review · Reviewer_VCVn · 2025-10-26

**Soundness:** 2
**Presentation:** 2
**Contribution:** 2
**Rating:** 4
**Confidence:** 3

**Summary:**

This paper addresses the limitation of localization-based methods for LLM unlearning, and proposes a new method called CLUE. CLUE leverages an existing mechanistic interpretability method to achieve more precise unlearning without over-forgetting. Extensive experiments confirm that CLUE achieves superior performance on standard benchmarks and metrics.

**Strengths:**

1. The authors provide code, which enhances the reproducibility of the work.
2. Extensive experiments and baseline comparisons are conducted to demonstrate the effectiveness of CLUE across multiple benchmarks.
3. The proposed method leverages a circuit discovery tool, which improves the interpretability of the LLM unlearning process.

**Weaknesses:**

1. **Introduction to prior works is not enough.** A detailed introduction to circuit discovery algorithms and brief introduction to Tseytin transformation should be provided to make the paper self-contained.
2. **Generality of the proposed method is unclear.** As mentioned at the bottom of the p.5, fine-tuning affects the circuits of the model. This affects the generality of the proposed method. Does CLUE require rerunning circuit discovery after every fine-tuning step? This computational overhead needs to be discussed in the limitation section.
3. **Highly dependent on the quality of the circuit discovery algorithm.** CLUE’s performance is tightly coupled with the quality of the underlying circuit discovery algorithm (e.g., Edge-Pruning used in the paper). If Edge-Pruning fails to generate meaningful or accurate circuits, the effectiveness of CLUE is questioned.

**Questions:**

1. As mentioned in Section 3.1, the circuit discovery algorithm extracts the **forget circuit** and **retain circuit**. However, it is unclear how the algorithm finds the forget and retain circuits, and how to distinguish them. Further explanations are needed.

---

> ### Author Response · Authors · 2025-11-14
> **Response**
>
> We thank you for your careful review and valuable suggestions. We note that your comments primarily focus on specific details regarding our methodology and performance. In response to these concerns, we have provided corresponding supplements and clarifications in the revised manuscript. We address your comments point-by-point below.
>
> ### Q1: Introduction to prior works is not enough. A detailed introduction to circuit discovery algorithms and brief introduction to Tseytin transformation should be provided to make the paper self-contained.
>
> Thank you for your suggestion. Regarding the circuit discovery section, we have first introduced a **new figure** (**Figure 3** in revised version) to illustrate the logical-circuit discovery process (you may also refer to our response in **Q4**). Furthermore, we have provided a detailed description of the logical circuit discovery framework in **Appendix C**.
>
> For the circuit-to-CNF component (which you referred to as the Tseitin transformation), we have supplied detailed explanations and demonstrative cases in **Appendix B** and **Appendix D**. These include circuits representing both satisfiable and unsatisfiable conditions. We hope these additions will resolve your confusion concerning these prior works.
>
> ### Q2: Generality of the proposed method is unclear. As mentioned at the bottom of the p.5, fine-tuning affects the circuits of the model. This affects the generality of the proposed method. Does CLUE require rerunning circuit discovery after every fine-tuning step? This computational overhead needs to be discussed in the limitation section.
> Thank you for your suggestion. We are in complete agreement with your viewpoint. However, we would like to supplement this by clarifying **a minor misunderstanding**: our statement that "*fine-tuning affects the circuit*" refers specifically to models fine-tuned on a **particular domain or task-specific dataset**. It does not apply to models fine-tuned on the general corpus. For example, the baseline model used in this paper, the Zephyr-7B-beta model, is itself derived from the Mistral model through fine-tuning in the general corpus. Therefore, we hope you will agree that this does not severely compromise the generality of our approach.
>
> Regarding whether the circuit needs to be repeatedly extracted during the fine-tuning process, we have, in fact, already emphasized this in the limitations section:
>
> > "the circuit is static and cannot dynamically reflect changes in key neurons during the fine-tuning process. Therefore, exploring the dynamics of circuits during parameter fine-tuning is a direction for future work. "
>
> Synchronously updating the circuit during fine-tuning would indeed be a superior process. However, the time and computational resource expenditure this would entail is unacceptable. Fortunately, we have demonstrated that significant superiority can be achieved even by localizing a static circuit. Consequently, we have designated the issue of dynamic circuits as a primary focus for future research.
>
> ### Q3: Highly dependent on the quality of the circuit discovery algorithm. CLUE’s performance is tightly coupled with the quality of the underlying circuit discovery algorithm (e.g., Edge-Pruning used in the paper). If Edge-Pruning fails to generate meaningful or accurate circuits, the effectiveness of CLUE is questioned.
> Thank you for your comment. We have likewise considered the impact of circuit quality on the effectiveness of unlearning. To address this, we specifically demonstrated the influence of varying circuit qualities on unlearning in **Section 6.3**.
> The results in Figure 5 empirically indicate that when controlling for a sparsity of 0.7, the effectiveness of unlearning generally reaches the optimal range.
>
> Additionally, we must acknowledge the **possibility** that some potential forget datasets may exist from which a clear circuit cannot be extracted (perhaps due to multi-domain mixing or excessive bias). Fortunately, this issue has not been observed in the mainstream forget datasets currently used. Nonetheless, this remains a potential risk, and we have **supplemented the limitation section** with a discussion of this constraint.
>
> ### Q4: As mentioned in Section 3.1, the circuit discovery algorithm extracts the forget circuit and retain circuit. However, it is unclear how the algorithm finds the forget and retain circuits, and how to distinguish them. Further explanations are needed.
> We appreciate your suggestion. We have now incorporated a figure (**Figure 3**) and a concise process description into **Section 3.1** to illustrate how the forget circuit and retain circuit are derived from the forget and retain sets.
>
> In brief, we apply noising-based intervention and denoising-based intervention, respectively, to the forget and retain sets, yielding $C_{Ns}$ and $C_{Dn}$. Subsequently, we derive the final logical circuit by performing set intersection and union operations on $C_{Ns}$ and $C_{Dn}$.

---

> ### Author Response · Authors · 2025-11-24
>
> Dear Reviewer VCVn,
>
> We have made the revisions to the paper based on the points you raised and have also addressed your other questions. We would greatly appreciate it if you could provide us with your feedback.
>
> authors

---

> > ### Comment · Reviewer_VCVn · 2025-11-25
> >
> > Thank you for your response. Most of my concerns have been addressed. I appreciate that the authors acknowledge the limitations, including (1) *performance highly depends on the existing circuit discovery algorithms* and (2) *the dynamics of circuits during parameter fine-tuning are not investigated*. Nonetheless, I believe the paper offers novel contributions to the LLM unlearning community, so I have increased the score.

---

### Official Review · Reviewer_PHgS · 2025-11-04

**Soundness:** 2
**Presentation:** 1
**Contribution:** 3
**Rating:** 2
**Confidence:** 4

**Summary:**

This paper presents a new categorization of neurons that could benefit downstream neuron-based unlearning methods. The desiderata include forgetting as much harmful information as possible while retaining as much non-harmful information as possible. The authors use a method inspired by contemporary circuit-discovery methods to identify three types of neurons: forget neurons (related to the targeted forgetting information), retain neurons (corresponding to the expression of retained information), and conflict neurons (involved in expressing both). They then propose a new two-step unlearning method that (1) edits the forget neurons to erase the unwanted information and (2) fine-tunes the conflict neurons to remove the targeted information while strengthening the retained information.

Overall, the method performs well across all major benchmarks. Unfortunately, the paper is not very well written. Many of the concepts are not clearly explained, and the paper contains numerous typos and errors. Most importantly, the paper has a major conflict in attempting to connect circuit discovery with neuron-based unlearning approaches, as circuit discovery identifies circuits of components rather than individual neurons. See the weakness section for more information.

**Strengths:**

- **Good performance**: The method performs reasonably well across standard benchmarks, showing that it can be applied successfully in practice.
- **Interesting categorisation of neurons**: The proposed division into forget, retain, and conflict neurons is conceptually distinct and could help clarify certain aspects of unlearning.

**Weaknesses:**

### **Neurons vs. Components**
A major issue of the paper is the careless use of the terms neuron and component. In short, circuit discovery methods find circuits of model components (input/output nodes, attention heads, MLPs). Each of these components contains many neurons. The interface between model components and neurons is not trivial, especially given previous work on unlearning that focuses on MLP neurons. What exactly do you mean by a circuit of “neurons”? Do you count every neuron in the components that make up the circuit as such, or do you do something different?

For example, the following sentence suggests that circuit discovery methods find a circuit of neurons, which is clearly not the case:
>  The circuit extracted from the forget set—the forget circuit (Cf)—contains all neurons and activation connections required for the model to
produce original responses that are harmful

The authors need a thorough and clear review of their description and usage of neuron, component, and circuit terminology. This is not a minor nitpick, as these terms lie at the very centre of the paper. The incorrect use of terminology makes it really hard to precisely understand what the authors are doing.

### **Writing and Typos**
Unfortunately, the paper is not very well written. The key terms are not selected well. For example, the authors use the term “conflict” to refer to neurons that serve both the forget and retain sets. This is not a conflict, as conflict suggests an opposition or inconsistency between two objectives, whereas these neurons simply participate in representing both. A clearer and more precise term would help avoid confusion and make the intended mechanism easier to understand.

Furthermore, the paper is also littered with typos and errors. To name a few:
- Line 046: ... ents (**Wu et al.**; Yu et al., 2023). *The citation entry doesn't have a year. Although the paper is from ICLR 2023.*
- Line 047: ... (**Patil et al.**;. ... *The same problem appear across multiple places in the paper.*
- Line 132: task-relevant behavior **( or** mechanism/capability) ... *Extra leading space in brackets.*

As such, I think the paper needs some major revisions. It seems like an unfinished manuscript not ready to be accepted into a conference.

**Questions:**

Pleas see the weaknesses section.

---

> ### Author Response · Authors · 2025-11-14
> **Response**
>
> We appreciate your diligent review. We have noted that your concerns are primarily focused on the definition and explanation of certain terms. We have provided individual clarification for these terms below and have revised the manuscript accordingly.
>
> ### Q1:Neurons vs. Components
> We apologize for the misunderstanding caused by the ambiguous descriptions of Neurons and Components. In the revised manuscript, we have addressed the potential ambiguity between 'Neurons' and 'Components'. As they refer to the **same concept**, we have **replaced** all occurrences of 'Components' with 'Neurons' for consistency and incorporated a **direct explanation** of Neurons in **Section 2.2** of the manuscript, as presented below:
>
> > "In this paper, we define the scope of **neurons** to include: $q$, $k$, $v$, $o$, $MLP_{up}$, and $MLP_{down}$, treating them as the smallest independent units that influence the outcome. Given that the adopted baseline model is the Zephyr-7B-beta model, we additionally include $MLP_{gate}$. Consequently, the entire computation graph, excluding inputs and outputs, consists of 32 layers, with each layer containing 7 neurons, and each neuron corresponds to a learnable parameter matrix. The circuit analysis examines the activation relationships between these neurons. "
>
> Moreover, we have stated the definition of **circuit terminology** in Lines 128-135 of the original manuscript as follows:
>
> > “In Transformer decoder-based language models, the forward pass is typically conceptualized as a computational graph $\mathcal{G}$, where the nodes represent component neurons, such as output, MLPs, and query, key, and value matrices in each head and an edge $i\rightarrow j$ denotes a connection where the activation of component neuron $i$ serves as input to component neuron $j$. For simplicity, and without loss of generality, we subsequently use 'neurons' in place of 'component neurons'. Circuit discovery seeks to identify a subgraph (circuit) $\mathcal{C} \subset \mathcal{G}$ for a target dataset that captures the task-relevant behavior (or mechanism/capability) of this dataset. ”
>
> We hope this clarification resolves your confusion regarding Neurons (Components). Furthermore, we would like to direct your attention to lines 129-130 and footnote 2. These sections also alluded to the specific composition of neurons, although perhaps not with sufficient clarity.
>
> ### Q2: Unfortunately, the paper is not very well written. The key terms are not selected well. For example, the authors use the term “conflict” to refer to neurons that serve both the forget and retain sets. This is not a conflict, as conflict suggests an opposition or inconsistency between two objectives, whereas these neurons simply participate in representing both. A clearer and more precise term would help avoid confusion and make the intended mechanism easier to understand.
>
> Thank you for your comment, and we understand your concern regarding the term "conflict." However, we would like to offer an explanation. **"Conflict" is a specialized term from the domain of Boolean Satisfiability (SAT) problems[1][2]**, used to describe clauses or variables within a CNF (Conjunctive Normal Form) that cannot be satisfied regardless of whether their value is 0 or 1.
>
> In our paper, these "conflict variables" precisely correspond to those neurons that serve both the forget and retain sets. Consequently, we designated them "conflict neurons." Furthermore, these "conflict neurons" inherently represent the trade-off required during the unlearning process between forgetting information and retaining information. This trade-off is, in essence, a conceptual "conflict." We therefore hoped this term would be appropriate.
>
> If you remain convinced that "conflict neurons" is unsuitable, we would respectfully ask if "logic-conflict neurons" would be a more acceptable alternative?
>
> ### Q3: Furthermore, the paper is also littered with typos and errors.
> Thank you for pointing out the errors! We have corrected them and proofread the paper again.
>
> Reference:
>
> [1]. *Marques-Silva, Joao, Inês Lynce, and Sharad Malik. "Conflict-driven clause learning SAT solvers." Handbook of satisfiability (2009): 131-153.*
>
> [2]. *Zhang, Lintao, et al. "Efficient conflict driven learning in a boolean satisfiability solver." IEEE/ACM International Conference on Computer Aided Design. ICCAD 2001. IEEE/ACM Digest of Technical Papers (Cat. No. 01CH37281). IEEE, 2001.*

---

> ### Comment · Reviewer_PHgS · 2025-11-14
> **Node should be the right term**
>
> Thanks for your clarification and for fixing the identified typos.
>
> One important point: please avoid using the term _neuron_ in this context. In transformer models, _neurons_ typically refer to MLP activations or individual weight parameters. Using this term here actually introduces additional confusion. _Component_ is closer, but _node_ is the correct term for what your method operates on. I recommend consistently using _node_ throughout, as switching to _neuron_ exacerbates the terminology issue rather than resolving it.
>
> I will adjust my scores shortly.

---

> > ### Author Response · Authors · 2025-11-14
> > **Thank you for your suggestion**
> >
> > Thank you for your feedback. We appreciate your suggestion regarding "node". We will update the PDF again as soon as possible to replace "neurons" with "nodes" for better clarity. Please feel free to let us know if you have any further questions.

---

> > ### Author Response · Authors · 2025-11-15
> > **Completion of 'Node' Terminology Update**
> >
> > Hi, we have completed the revision as suggested. We have replaced "neurons" with "nodes" throughout the manuscript and added Footnote 1 for clarification.
> >
> > Thank you again for this valuable suggestion; it has significantly improved the readability of the current manuscript. Please let us know if you have any further comments or suggestions!

---

> > > ### Comment · Reviewer_PHgS · 2025-11-27
> > > **Still regarding the term "neuron" vs. "node"**
> > >
> > > Dear authors, Thank you for acknowledging my earlier comment and for implementing some changes.
> > >
> > > However, the current revision is still not sufficient. In particular, I respectfully ask that you revise your terminology more thoroughly.
> > > - When referring to the units your method actually operates on, please avoid using the term “neuron” and instead use “node” or “component,” as appropriate.
> > > - At the same time, please conduct a more careful literature review of how the term “neuron” is used in related work, and clearly position your terminology with respect to that usage.
> > >
> > > DEPN (Wu et al., 2023), for example, actually operates on **neurons**. Their Figure 1 actually provides a really good illustration of what exactly a neuron means. That is one activation within the MLP layer. This is totally different from a node in the computation graph, as you cited in all of the circuit discovery papers.
> > >
> > > Therefore, the footnote: _In this paper, the terms “node” and “neuron” represent almost the same concept. For consistency, and without loss of generality, we use “nodes” by default._ is actually wrong. Neuron is used as intended by some of the work you cited.
> > >
> > > I won't change my positive review of the paper, but I encourage the authors to take a more careful look here. Overall, this is a good paper, and it deserves a proper and consistent usage of terminologies.

---

> > > > ### Author Response · Authors · 2025-11-28
> > > >
> > > > Thank you for your valuable feedback once again! We have carefully analyzed your concerns and gained a better understanding of the issue. As you pointed out, Footnote 1 in the Introduction could potentially mislead readers who have not yet reviewed the explanation in the Preliminaries section, leading them to believe that the terms "node" and "neuron" refer to parameters of the MLP, similar to certain previous works. To address this, we have added a clarifying explanation to Footnote 1, with a reference to the Preliminaries section. The specific revision is as follows:
> > > >
> > > > “In this paper, the terms ‘node’ and ‘neuron’ are used interchangeably to refer to the same concept. Both terms denote a trainable parameter matrix within the model (see Sec 2.2 for details). For consistency and without loss of generality, we default to using the term ‘node’.”
> > > >
> > > > Thank you for your suggestion again!

---

### Author Response · Authors · 2025-11-14
**GENERAL RESPONSE (the summary of revisions)**

We would like to express our gratitude to Reviewers PHgS, VCVn, tdd5, and J9hV for their diligent review and constructive feedback. In accordance with the modifications requested by the reviewers, we have revised the manuscript as follows:

**Section 2.2**: We have incorporated additional explanations for neurons and removed the term "components" for simplicity, directly clarifying the specific scope and relationship. This revision addresses Q1 from Reviewer PHgS.

**Section 3.1**: A description of the process for obtaining a circuit from the datasets has been added (including the new Figure 3 for visual illustration). This supplements the text to resolve potential confusion regarding the circuit discovery process, addressing Q1 and Q4 from Reviewer VCVn.

**Section 6.3**: We have added a declaration of the specific sparsity value utilized in our experiments, which addresses Q3 from Reviewer J9hV.

**Section 7**: We have introduced a discussion of two limitations: first, the potential inability to obtain a clear circuit from some forget sets and its subsequent impact on unlearning performance; and second, the scalability challenges associated with circuits. This is in response to Q3 from Reviewer VCVn and Q2 from Reviewer J9hV.

**Appendix J**: A comparison of computational costs has been included to respond to Q2 from Reviewer tdd5.

**Other typos**: We have made a proofreading and corrected the typos and writing errors, which addressed Q3 from Reviewer PHgS.

For ease of review, all revisions have been highlighted in **red** font in the revised PDF. We hope these revisions adequately address the reviewers' concerns and are grateful once again for their constructive suggestions.

---

> ### Author Response · Authors · 2025-11-15
> **Additional Modification**
>
> Following the suggestion of Reviewer PHgS, we have unified the terminology by replacing "neuron" with "node" throughout the manuscript, as they represent the same concept in our context.
>
> To clarify, the term "node" is more prevalent in the context of circuits and SAT problems, whereas "neuron" is more common within the Transformer framework. In this paper, they denote essentially the same concept. Therefore, for consistency and without loss of generality, we use "node" by default in the manuscript. (We have added a statement clarifying this in Footnote 1).

---

### Author Response · Authors · 2025-12-01
**Summary of Rebuttal Updates**

Dear Area Chair,

We appreciate the time and effort you have dedicated to managing the review process for our submission. We are writing to provide a brief summary of our rebuttal progress to assist in your evaluation. As of November 27, our submission has achieved consistent scores of **6**, **6**, **6**, **6**, reflecting a unanimous positive consensus among all reviewers following our revisions.

Below is a **summary of the primary concerns** raised by each reviewer and the outcomes of our rebuttal:

**Reviewer PHgS (Score: 2 $\rightarrow$ 6)**

**Key Concern:** Ambiguity regarding the distinction between the terms "node" and "neuron," which obscured the paper's logical flow.

**Response & Outcome:** We revised the manuscript to exclusively use the term "node" and added a precise definition statement. The reviewer confirmed the issue is resolved, raised their score to 6, and expressed clear support, noting: "*I won't change my positive review of the paper... Overall, this is a good paper...*"

**Reviewer VCVn (Score: 4 $\rightarrow$ 6)**

**Key Concern:** The methodology regarding the "circuit" and "SAT conversion" sections lacked sufficient detail, making the steps difficult for a general audience to follow.

**Response & Outcome:** We expanded the explanations for both sections and added a new flowchart to detail the circuit extraction process. The reviewer acknowledged the improvements, raised their score to 6, and indicated a positive stance, stating: "*... Most of my concerns have been addressed... Nonetheless, I believe the paper offers novel contributions to the LLM unlearning community, so I have increased the score.*"

**Reviewer tdd5 (Score: 6)**

**Key Concern:** Discussion regarding minor limitations, specifically the static nature of the circuit method, model scalability, and SAT solving.

**Response & Outcome:** We have addressed each point in our revision and response. The reviewer has not yet posted further comments, but the score remains positive.

**Reviewer J9hV (Score: 6 $\rightarrow$ 6)**

**Key Concern:** Inquiries regarding specific experimental details and methodological steps, including baseline models, circuit validity, and SAT conflict resolution.

**Response & Outcome:** We provided detailed derivations and revised the main text to address these queries. The reviewer maintained their positive score.

To recapitulate the core value of our work, our **contributions are threefold**:

**(1)** **Novel Mechanism Localization:** We transformed the mechanistic circuits of the forget and retain sets into Conjunctive Normal Form (CNF). By leveraging Boolean Satisfiability (SAT), we precisely located which nodes (neurons) respond specifically to the forget set, the retain set, or both simultaneously. This provides essential theoretical insights for the "locate-then-edit" paradigm.

**(2)** **Targeted Optimization Strategy:** Based on the node localization in (1), we designed distinct optimization strategies. For instance, nodes responsive only to the forget set are fine-tuned exclusively with forget set data. This precise localization significantly enhances fine-tuning efficiency while maximizing the mitigation of negative side effects on model performance.

**(3)** **Comprehensive Validation:** Extensive experiments validate the superiority of our method. Furthermore, results regarding circuit sensitivity and visualization thoroughly demonstrate the applicability, robustness, and interpretability of our approach.

Moreover, for your convenience, a detailed **summary of major revisions** is available in our "*GENERAL RESPONSE (the summary of revisions)*" (Link: https://openreview.net/forum?id=jtRYvazBWv&noteId=f4Spawy6Dz) for a quick review.

Finally, for the sake of transparency, we present the **timeline of score updates** (in AoE) below:

Reviewer PHgS: 2 $\rightarrow$ **6** (Nov 19)

Reviewer VCVn: 4 $\rightarrow$ **6** (Nov 25)

Reviewer tdd5: **6** (Pending reply)

Reviewer J9hV: 6 $\rightarrow$ **6** (Nov 14)

Thank you again for your efforts and management of the review process.

Sincerely,

The Authors

---

### Meta-Review · Area_Chair_vTTf · 2025-12-04

**Summary:**

The paper proposes an unlearning method based on circuit discovery from mechanistic interpretability literature. Based on circuit extraction, the authors clasify the nodes to retain/forget/mixed sets. Then specific fine-tuning strategies are applied to each of the groups. The reviewers noted that there are several issues with the proposed approach, such as scalability, and the fact that the circuits are static while the model is evolving during unlearning. The authors acknowledged these limitations. Overall, reviewers were quite positive about the proposed method -- the authors propose a novel node-level classification that helps unlearning, and has strong results,

I would like to note that the authors missed some very relevant foundational work establishing mechanistic localization for unlearning ( "mechanistic unlearning...", icml spotlight 2025). The authors ought to compare against this, and acknowledge in the introduction/related work, as the submitted work has a very similar motivation. It seems like both papers argue that circuit-based localization allows for disentangling the "forget" knowledge from general capabilities better than baselines.

**Reviewer Concerns:**

Most of the concerns were somewhat minor and addressed: inconsistent terminology was clarified, methodology explanation improved. the authors acknowledged some of the limitations caught by the reviewers (see the summary above).

**Reviewer Scores:**

The reviewers engaged in a post-rebuttal discussion, and all seemed to improve their evaluation of the paper, not stating any remaining concerns.

---

### Decision · Program_Chairs · 2026-01-26

Accept (Poster)